# PROACTIVE MULTI-CAMERA COLLABORATION FOR 3D HUMAN POSE ESTIMATION

**Hai Ci**[*†§]**, Mickel Liu**[*†§]**, Xuehai Pan**[*†§]**, Fangwei Zhong**[✉◇§]**, Yizhou Wang**[†§♡♠]

[†] School of Computer Science, Peking University
[§] Nat'l Key Lab. of GAI & Beijing Institute for GAI (BIGAI)
[◇] School of Intelligence Science and Technology, Peking University
[♡] Inst. for AI, Peking University     [♠] Nat'l Eng. Research Center of Visual Technology
{cihai, XuehaiPan, zfw, yizhou.wang}@pku.edu.cn, mickelliu@stu.pku.edu.cn

## ABSTRACT

This paper presents a multi-agent reinforcement learning (MARL) scheme for proactive Multi-Camera Collaboration in 3D Human Pose Estimation in dynamic human crowds. Traditional fixed-viewpoint multi-camera solutions for human motion capture (MoCap) are limited in capture space and susceptible to dynamic occlusions. Active camera approaches proactively control camera poses to find optimal viewpoints for 3D reconstruction. However, current methods still face challenges with credit assignment and environment dynamics. To address these issues, our proposed method introduces a novel Collaborative Triangulation Contribution Reward (CTCR) that improves convergence and alleviates multi-agent credit assignment issues resulting from using 3D reconstruction accuracy as the shared reward. Additionally, we jointly train our model with multiple world dynamics learning tasks to better capture environment dynamics and encourage anticipatory behaviors for occlusion avoidance. We evaluate our proposed method in four photo-realistic UE4 environments to ensure validity and generalizability. Empirical results show that our method outperforms fixed and active baselines in various scenarios with different numbers of cameras and humans.

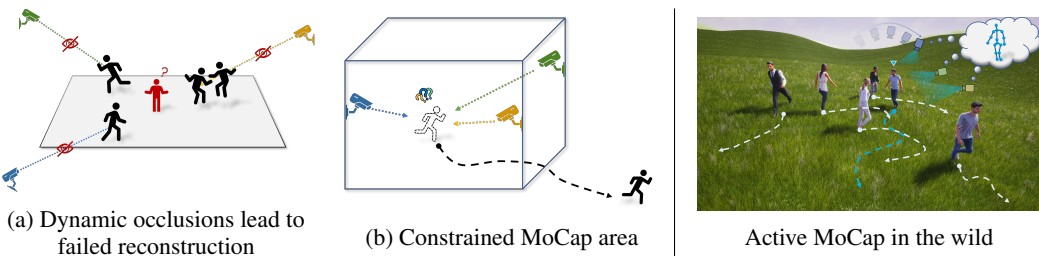

(a) Dynamic occlusions lead to failed reconstruction     (b) Constrained MoCap area     Active MoCap in the wild

Figure 1: **Left**: Two critical challenges in fixed camera approaches. **Right**: Three active cameras collaborate to best reconstruct the 3D pose of the target (marked in ▼).

## 1 INTRODUCTION

Marker-less motion capture (MoCap) has broad applications in many areas such as cinematography, medical research, virtual reality (VR), sports, and *etc*. Their successes can be partly attributed to recent developments in 3D Human pose estimation (HPE) techniques (Tu et al., 2020; Iskakov et al., 2019; Jafarian et al., 2019; Pavlakos et al., 2017b; Lin & Lee, 2021b). A straightforward implementation to solve multi-views 3D HPE is to use fixed cameras. Although being a convenient solution, it is less effective against dynamic occlusions. Moreover, fixed camera solutions confine tracking targets within a constrained space, therefore less applicable to outdoor MoCap. On the contrary, active cameras (Luo et al., 2018; 2019; Zhong et al., 2018a; 2019) such as ones mounted on drones can maneuver proactively against incoming occlusions. Owing to its remarkable flexibility, the active approach has thus attracted overwhelming interest (Tallamraju et al., 2020; Ho et al., 2021; Xu et al., 2017; Kiciroglu et al., 2019; Saini et al., 2022; Cheng et al., 2018; Zhang et al., 2021).

---

[*]Equal Contribution. ✉Corresponding author.
    Project Website: https://sites.google.com/view/active3dpose

Previous works have demonstrated the effectiveness of using active cameras for 3D HPE on a single target in indoor (Kiciroglu et al., 2019; Cheng et al., 2018), clean landscapes (Tallamraju et al., 2020; Nägeli et al., 2018; Zhou et al., 2018; Saini et al., 2022) or landscapes with scattered static obstacles (Ho et al., 2021). However, to the best of our knowledge, we have not seen any existing work that experimented with *multiple* ($n > 3$) active cameras to conduct 3D HPE *in human crowd*. There are two key challenges : **First**, frequent human-to-human interactions lead to **random dynamic occlusions**. Unlike previous works that only consider clean landscapes or static obstacles, dynamic scenes require frequent adjustments of cameras' viewpoints for occlusion avoidance while keeping a good overall team formation to ensure accurate multi-view reconstruction. Therefore, achieving optimality in dynamic scenes by implementing a fixed camera formation or a hand-crafted control policy is challenging. In addition, the complex behavioural pattern of a human crowd makes the occlusion patterns less comprehensible and predictable, further increasing the difficulty in control. **Second**, as the team size grows larger, the **multi-agent credit assignment issue** becomes prominent which hinders policy learning of the camera agents. Concretely, multi-view 3D HPE as a team effort requires inputs from multiple cameras to generate an accurate reconstruction. Having more camera agents participate in a reconstruction certainly introduces more redundancy, which reduces the susceptibility to reconstruction failure caused by dynamic occlusions. However, it consequently weakens the association between individual performance and the reconstruction accuracy of the team, which leads to the "lazy agent" problem (Sunehag et al., 2017).

In this work, we introduce a proactive multi-camera collaboration framework based on multi-agent reinforcement learning (MARL) for real-time distributive adjustments of multi-camera formation for 3D HPE in a human crowd. In our approach, multiple camera agents perform seamless collaboration for successful reconstructions of 3D human poses. Additionally, it is a decentralized framework that offers flexibility over the formation size and eliminates dependency on a control hierarchy or a centralized entity. Regarding the **first** challenge, we argue that the model's ability to predict human movements and environmental changes is crucial. Thus, we incorporate *World Dynamics Learning (WDL)* to train a state representation with these properties, *i.e.*, learning with five auxiliary tasks to predict the target's position, pedestrians' positions, self state, teammates' states, and team reward.

To tackle the **second** challenge, we further introduce the *Collaborative Triangulation Contribution Reward (CTCR)*, which incentivizes each agent according to its characteristic contribution to a 3D reconstruction. Inspired by the Shapley Value (Rapoport, 1970), CTCR computes the average weighted marginal contribution to the 3D reconstruction for any given agent over all possible coalitions that contain it. This reward aims to directly associate agents' levels of participation with their adjusted return, guiding their policy learning when the team reward alone is insufficient to produce such direct association. Moreover, CTCR penalizes occluded camera agents more efficiently than the shared reward, encouraging emergent occlusion avoidance behaviors. Empirical results show that CTCR can accelerate convergence and increase reconstruction accuracy. Furthermore, CTCR is a general approach that can benefit policy learning in active 3D HPE and serve as a new assessment metric for view selection in other multi-view reconstruction tasks.

For the evaluations of the learned policies, we build photo-realistic environments (*UnrealPose*) using Unreal Engine 4 (UE4) and UnrealCV (Qiu et al., 2017). These environments can simulate realistic-behaving crowds with assurances of high fidelity and customizability. We train the agents on a *Blank* environment and validate their policies on three unseen scenarios with different landscapes, levels of illumination, human appearances, and various quantities of cameras and humans. The empirical results show that our method can achieve more accurate and stable 3D pose estimates than off-the-shelf passive- and active-camera baselines. To help facilitate more fruitful research on this topic, we release our environments with OpenAI Gym-API (Brockman et al., 2016) integration and together with a dedicated visualization tool.

Here we summarize the key contributions of our work:

- Formulating the active multi-camera 3D human pose estimation problem as a Dec-POMDP and proposing a novel multi-camera collaboration framework based on MARL (with $n \geq 3$).
- Introducing five auxiliary tasks to enhance the model's ability to learn the dynamics of highly dynamic scenes.
- Proposing CTCR to address the credit assignment problem in MARL and demonstrating notable improvements in reconstruction accuracy compared to both passive and active baselines.
- Contributing high-fidelity environments for simulating realistic-looking human crowds with authentic behaviors, along with visualization software for frame-by-frame video analysis.

## 2 RELATED WORK

**3D Human Pose Estimation (HPE)** Recent research on 3D human pose estimation has shown significant progress in recovering poses from single monocular images (Ma et al., 2021; Pavlakos et al., 2017a; Martinez et al., 2017; Kanazawa et al., 2018; Pavlakos et al., 2018; Sun et al., 2018; Ci et al., 2019; Zeng et al., 2020; Ci et al., 2020) or monocular video (Mehta et al., 2017; Hossain & Little, 2018; Pavllo et al., 2019; Kocabas et al., 2020). Other approaches utilize multi-camera systems for triangulation to improve visibility and eliminate ambiguity (Qiu et al., 2019; Jafarian et al., 2019; Pavlakos et al., 2017b; Dong et al., 2019; Lin & Lee, 2021b; Tu et al., 2020; Iskakov et al., 2019). However, these methods are often limited to indoor laboratory environments with fixed cameras. In contrast, our work proposes an active camera system with multiple mobile cameras for outdoor scenes, providing greater flexibility and adaptability.

**Proactive Motion Capture** Few previous works have studied proactive motion capture with a single mobile camera (Zhou et al., 2018; Cheng et al., 2018; Kiciroglu et al., 2019). In comparison, more works have studied the control of a multi-camera team. Among them, many are based on optimization with various system designs, including marker-based (Nägeli et al., 2018), RGBD-based (Xu et al., 2017), two-stage system (Saini et al., 2019; Tallamraju et al., 2019), hierarchical system (Ho et al., 2021), *etc*. It is important to note that all the above methods deal with static occlusion sources or clean landscapes. Additionally, the majority of these works adopt hand-crafted optimization objectives and some forms of fixed camera formations. These factors result in poor adaptability to dynamic scenes that are saturated with uncertainties. Recently, RL-based methods have received more attention due to their potential for dynamic formation adjustments. These works have studied active 3D HPE in the Gazebo simulation (Tallamraju et al., 2020) or Panoptic dome (Joo et al., 2015; Pirinen et al., 2019; Gärtner et al., 2020) for active view selection. Among them, AirCapRL (Tallamraju et al., 2020) shares similarities with our work. However, it is restricted to coordinating between two cameras in clean landscapes without occlusions. We study collaborations between multiple cameras ($n \geq 3$) and resolve the credit assignment issue with our novel reward design (CTCR). Meanwhile, we study a more challenging scenario with multiple distracting humans serving as sources of dynamic occlusions, which requires more sophisticated algorithms to handle.

**Multi-Camera Collaboration** Many works in computer vision have studied multi-camera collaboration and designed active camera systems accordingly. Earlier works (Collins et al., 2003; Qureshi & Terzopoulos, 2007; Matsuyama et al., 2012) focused on developing a network of pan-tile-zoom (PTZ) cameras. Owing to recent advances in MARL algorithms (Lowe et al., 2017; Sunehag et al., 2017; Rashid et al., 2018; Wang et al., 2020; Yu et al., 2021; Jin et al., 2022), many works have formulated multi-camera collaboration as a multi-agent learning problem and solved it using MARL algorithms accordingly (Li et al., 2020; Xu et al., 2020; Wenhong et al., 2022; Fang et al., 2022; Sharma et al., 2022; Pan et al., 2022). However, most works focus on the target tracking problem, whereas this work attempts to solve the task of 3D HPE. Compared with the tracking task, 3D HPE has stricter criteria for optimal view selections due to correlations across multiple views, which necessitates intelligent collaboration between cameras. To our best knowledge, this work is the first to experiment with various camera agents ($n \geq 3$) to learn multi-camera collaboration strategies for active 3D HPE.

## 3 PROACTIVE MULTI-CAMERA COLLABORATION

This section will explain the formulation of multi-camera collaboration in 3D HPE as a Dec-POMDP. Then, we will describe our proposed solutions for modelling the virtual environment's complex dynamics and strengthening credit assignment in the multi-camera collaboration task.

### 3.1 PROBLEM FORMULATION

We formulate the multi-camera 3D HPE problem as a Decentralized Partially-Observable Markov Decision Process (Dec-POMDP), where each camera is considered as an agent which is decentralized-controlled and has partial observability over the environment. Formally, a Dec-POMDP is defined as $\langle \mathcal{S}, \mathcal{O}, \mathcal{A}^n, n, P, r, \gamma \rangle$, where $\mathcal{S}$ denotes the global state space of the environment, including all human states and camera states in our problem. $o_i \in \mathcal{O}$ denotes the agent $i$'s local observation, *i.e.*, the RGB image observed by camera $i$. $\mathcal{A}$ denotes the action space of an agent and $\mathcal{A}^n$ represents the joint action space of all $n$ agents. $P : \mathcal{S} \times \mathcal{A}^n \rightarrow \mathcal{S}$ is the transition probability function $P(s^{t+1}|s^t, \boldsymbol{a}^t)$, in which $\boldsymbol{a}^t \in \mathcal{A}^n$ is a joint action by all $n$ agents. At each timestep $t$, every agent obtains a local view $o_i^t$ from the environment $s^t$ and then preprocess $o_i^t$ to form $i$-th agent's local

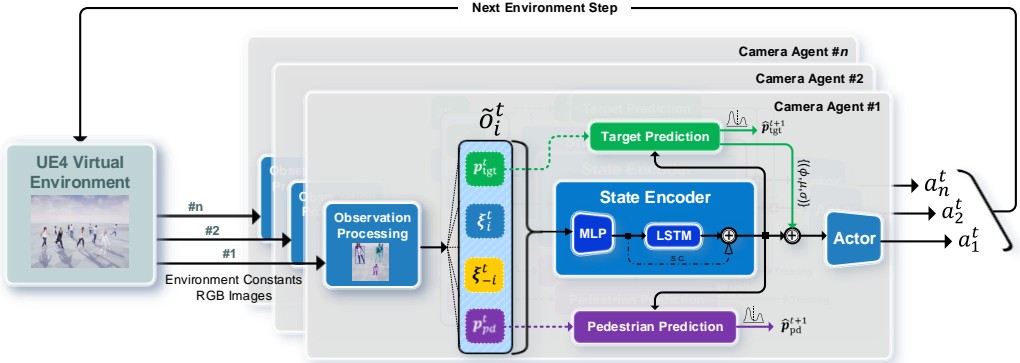

Figure 2: Simplified Pipeline Architecture. For $i$ agent at $t$ step, the environment returns local camera view $o_i^t$ and environment constants that are pre-processed to generate local observation $\tilde{o}_i^t$ (described here). Target Prediction and Pedestrian Prediction modules generate predictions $\hat{p}_{\text{tgt}}^t$ and $\hat{p}_{\text{pd}}^t$, respectively. The parameters of the Mixture Density Network(MDN) within the Target Prediction Module are concatenated to the state encoder's output to improve the quality of the representation.

observation $\tilde{o}_i^t$. The agent performs action $a_i^t \sim \pi_i^t(\cdot|\tilde{o}_i^t)$ and receives its reward $r(s^t, \boldsymbol{a}^t)$. $\gamma \in (0, 1]$ is the discount factor used to calculate the cumulative discounted reward $G(t) = \sum_{t' \geq t} \gamma^{t'-t} r(t')$. In a cooperative team, the objective is to learn a group of decentralized policies $\{\pi_i(a_i^t|\tilde{o}_i^t)\}_{i=1}^n$ that maximizes $\mathbb{E}_{(s, \boldsymbol{a}) \sim \boldsymbol{\pi}}[G(t)]$. For convenience, we denote $i$ as the agent index, $[\![n]\!] = \{1, \dots, n\}$ is the set of all $n$ agents, and $-\boldsymbol{i} = [\![n]\!] \setminus \{i\}$ are all $n$ agents except agent $i$.

**Observation**   Camera agents have partial observability over the environment. The pre-processed observation $\tilde{o}_i = (p_i, \xi_i, \xi_{-\boldsymbol{i}})$ of the camera agent $i$ consists of: **(1)** $p_i$, a set of states of visible humans to agent $i$, containing information, including the detected human bounding-box in the 2D local view, the 3D positions and orientations of all visible humans measured in both local coordinate frame of camera $i$ and world coordinates; **(2)** own camera pose $\xi_i$ showing the camera's position and orientation in world coordinates; **(3)** peer cameras poses $\xi_{-\boldsymbol{i}}$ showing their positions and orientations in world coordinates and are obtained via multi-agent communication.

**Action Space**   The action space of each camera agent consists of the velocity of 3D egocentric translation $(x, y, z)$ and the velocity of 2D pitch-yaw rotation $(\theta, \psi)$. To reduce the exploration space for state-action mapping, the agent's action space is discretized into three levels across all five dimensions. At each timestep, the camera agent can move its position by $[+\delta, 0, -\delta]$ in $(x, y, z)$ directions and rotate about pitch-yaw axes by $[+\eta, 0, -\eta]$ degrees. In our experiments, the camera's pitch-yaw angles are controlled by a rule-based system.

### 3.2   FRAMEWORK

This section will describe the technical framework that constitutes our camera agents, which contains a Perception Module and a Controller Module. The Perception Module maps the original RGB images taken by the camera to numerical observations. The Controller Module takes these numerical observations and produces corresponding control signals. Fig. 2 illustrates this framework.

**Perception Module**   The perception module executes a procedure consisting of four sequential stages: **(1)** 2D HPE. The agent performs 2D human detection and pose estimation on the observed RGB image with the YOLOv3 (Redmon & Farhadi, 2018) detector and the HRNet-w32 (Sun et al., 2019) pose estimator, respectively. Both models are pre-trained on the COCO dataset, (Lin et al., 2014) and their parameters are kept frozen during policy learning of camera agents to ensure cross-scene generalization. **(2)** Person ReID. A ReID model (Zhong et al., 2018c) is used to distinguish people in a scene. For simplicity, an appearance dictionary of all to-be-appeared people is built in advance following (Gärtner et al., 2020). At test time, the ReID network computes features for all detected people and identifies different people by comparing features to the pre-built appearance dictionary. **(3)** Multi-agent Communication. Detected 2D human pose, IDs, and own camera pose are broadcasted to other agents. **(4)** 3D HPE. 3D human pose is reconstructed via local triangulation after receiving communications from other agents. The estimated position and orientation of a person can then be extracted from the corresponding reconstructed human pose. The communication process is illustrated in Appendix Fig. 9.

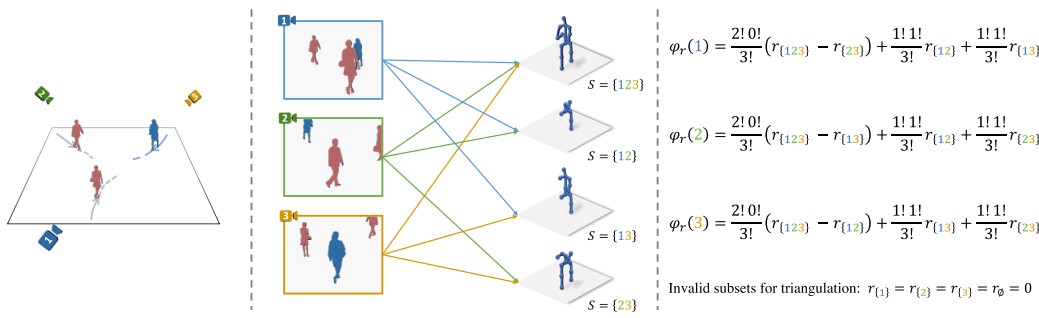

Figure 3: An instance of computing Collaborative Triangulation Contribution Reward (CTCR) for every camera. **Left:** Three cameras tasked to perform HPE on the target person avoiding occlusions from two pedestrians. **Middle:** Three cameras can constitute four valid coalitions, each generating a different 3D HPE. **Right:** Compute CTCR (without re-scaling by $n$) for each camera.

**Controller Module**   The controller module consists of a state encoder $E$ and an actor network $A$. The state encoder $E$ takes $\tilde{o}_i^t$ as input, encoding the temporal dynamics of the environment via LSTM. The future states of the target, pedestrians, and cameras are modelled using Mixture Density Network (MDN) (Bishop, 1994) to account for uncertainty. During model inference, it computes target position prediction, and then the $(\phi, \mu, \sigma)$ parameters of target prediction MDN are used as a part of the inputs to the actor network to enhance feature encoding. Please refer to Section 3.4 for more details regarding training the MDN.

$$\text{Feature Embedding} \quad z_i^t = \text{MLP}(\tilde{o}_i^t), \tag{1}$$

$$\text{Temporal Modeling} \quad h_i^t = \text{LSTM}(z_i^t, h_i^{t-1}), \tag{2}$$

$$\text{Human Trajectory Prediction} \quad \hat{p}_{\text{tgt/pd}}^{t+1} = \text{MDN}(z_i^t, h_i^t, p_{\text{tgt/pd}}^t), \tag{3}$$

$$\text{Final Embedding} \quad e_i^t = E\left(\tilde{o}_i^t, h_i^{t-1}\right) = \text{Concat}\left(z_i^t, h_i^t, \{(\phi, \mu, \sigma)\}_{\text{MDN}_{\text{tgt}}}\right), \tag{4}$$

where $p_{\text{tgt}}$ and $p_{\text{pd}}$ refer to the state of the target and the pedestrian, respectively. The actor network $A$ consists of 2 fully-connected layers that output the action, $a_i^t = A(E(\tilde{o}_i^t, h_i^{t-1}))$.

## 3.3   REWARD STRUCTURE

To alleviate the credit assignment issue that arises in multi-camera collaboration, we propose the Collaborative Triangulation Contribution Reward (CTCR). We start by defining a base reward that reflects the reconstruction accuracy of the triangulated pose generated by the camera team. Then we explain how our CTCR is computed based on this base team reward.

**Reconstruction Accuracy as a Team Reward**   To directly reflect the reconstruction accuracy, the reward function negatively correlates with the pose estimation error (Mean Per Joint Position Error, MPJPE) of the multi-camera triangulation. Formally,

$$r(X) = \begin{cases} 0, & |X| \leq 1, \\ 1 - \text{Gemen}\left(\text{MPJPE}(X)\right), & |X| \geq 2. \end{cases} \tag{5}$$

Where the set $X$ represents the participating cameras in triangulation, and employs the Geman-McClure smoothing function, $\text{Gemen}(\cdot) = \frac{2(\cdot/c)^2}{(\cdot/c)^2 + 4}$, to stabilize policy updates, where $c = 50\text{mm}$ in our experiments. However, the shared team reward structure in our MAPPO baseline, where each camera in the entire camera team $X$ receives a common reward $r(X)$, presents a credit assignment challenge, especially when a camera is occluded, resulting in a reduced reward for all cameras. To address this issue, we propose a new approach called Collaborative Triangulation Contribution Reward (CTCR).

**Collaborative Triangulation Contribution Reward (CTCR)**   CTCR computes each agent's individual reward based on its marginal contribution to the collaborative multi-view triangulation. Refer to Fig. 3 for a rundown of computing CTCR for a 3-cameras team. The contribution of agent $i$ can be measured by:

$$\text{CTCR}(i) = n \cdot \varphi_r(i), \qquad \varphi_r(i) = \sum_{S \subseteq \llbracket n \rrbracket \setminus \{i\}} \frac{|S|!(n - |S| - 1)!}{n!}[r(S \cup \{i\}) - r(S)], \tag{6}$$

Where $n$ denotes the total number of agents. $S$ denotes all the subsets of $[\![n]\!]$ not containing agent $i$. $\frac{|S|!(n-|S|-1)!}{n!}$ is the normalization term. $[r(S \cup \{i\}) - r(S)]$ means the marginal contribution of agent $i$. Note that $\sum_{i \in [\![n]\!]} \varphi_r(i) = r([\![n]\!])$. We additionally multiply a constant $n$ to rescale the CTCR to have the same scale as the team reward. Especially in the 2-cameras case, the individual CTCR should be equivalent to the team reward, *i.e.*, $\text{CTCR}(i=1) = \text{CTCR}(i=2) = r(\{1,2\})$. For more explanations on CTCR, please refer to Appendix Section G.

### 3.4 LEARNING MULTI-CAMERA COLLABORATION VIA MARL

We employ the multi-agent learning variant of PPO (Schulman et al., 2017) called Multi-Agent PPO (MAPPO) (Yu et al., 2021) to learn the collaboration strategy. Alongside the RL loss, we jointly train the model with five auxiliary tasks that encourage comprehension of the world dynamics and the stochasticity in human behaviours. The pseudocode can be found in Appendix A.

**World Dynamics Learning (WDL)** We use the encoder's hidden states $(z_i^t, h_i^t)$ as the basis to model the world. Three WDL objectives correspond to modelling agent dynamics : **(1)** learning the forward dynamics of the camera $P_1(\xi_i^{t+1}|z_i^t, h_i^t, a_i^t)$, **(2)** prediction of team reward $P_2(r^t|z_i^t, h_i^t, a_i^t)$, **(3)** prediction of future position of peer agents $P_3(\xi_{-i}^{t+1}|z_i^t, h_i^t, a_i^t)$. Two WDL objectives correspond to modelling human dynamics: **(4)** prediction of future position of target person $P_4(p_{\text{tgt}}^{t+1}|z_i^t, h_i^t, p_{\text{tgt}}^t)$, **(5)** prediction of future position of pedestrians, $P_5(p_{\text{pd}}^{t+1}|z_i^t, h_i^t, p_{\text{pd}}^t)$. All the probability functions above are approximated using Mixture Density Networks (MDNs) (Bishop, 1994).

**Total Training Objectives** $\mathcal{L}_{\text{Train}} = \mathcal{L}_{\text{RL}} + \lambda_{\text{WDL}} \mathcal{L}_{\text{WDL}}$. The $\mathcal{L}_{\text{RL}}$ is the reinforcement learning loss consisting of PPO-Clipped loss and centralized-critic network loss similar to MAPPO (Yu et al., 2021). $\mathcal{L}_{\text{WDL}} = -\frac{1}{n} \sum_l \lambda_l \sum_i \mathbb{E}[\log P_l(\cdot|\tilde{o}_i^t, h_i^t, a_i^t)]$ is the world dynamics learning loss that consists of MDN supervised losses on the five prediction tasks mentioned above.

## 4 EXPERIMENT

In this section, we first introduce our novel environment, UNREALPOSE, used for training and testing the learned policies. Then we compare our method with multi-passive-camera baselines and perform an ablation study on the effectiveness of the proposed CTCR and WDL objectives. Additionally, we evaluate the effectiveness of the learned policies by comparing them against other active multi-camera methods. Lastly, we test our method in four different scenarios to showcase its robustness.

### 4.1 UNREALPOSE: A VIRTUAL ENVIRONMENT FOR PROACTIVE HUMAN POSE ESTIMATION

We built four virtual environments for simulating active HPE in the wild using Unreal Engine 4 (UE4), which is a powerful 3D game engine that can provide real-time and photo-realistic renderings for making visually-stunning video games. The environments handle the interactions between realistic-behaving human crowds and camera agents. Here is a list of characteristics of UNREALPOSE that we would like to highlight: **Realistic**: Diverse generations of human trajectories, built-in collision avoidance, and several scenarios with different human appearance, terrain, and level of illumination. **Flexibility**: extensive configuration in numbers of humans, cameras, or their physical properties, more than 100 MoCap action sequences incorporated. **RL-Ready**: integrated with OpenAI Gym API, overhaul the communication module in the UnrealCV (Qiu et al., 2017) plugin with inter-process communication (IPC) mechanism. For more detailed descriptions, please refer to Appendix Section B.

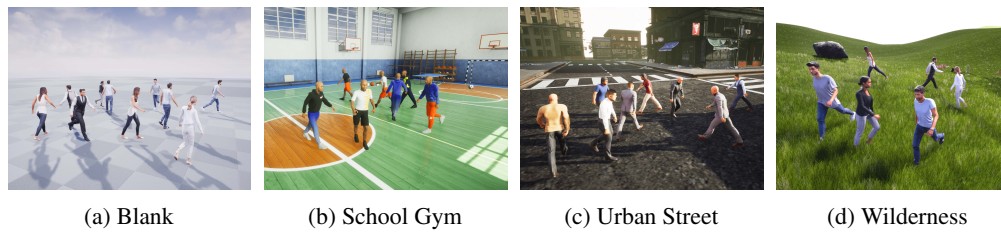

(a) Blank        (b) School Gym        (c) Urban Street        (d) Wilderness

Figure 4: UNREALPOSE: Our virtual environments for active 3D HPE in human crowd

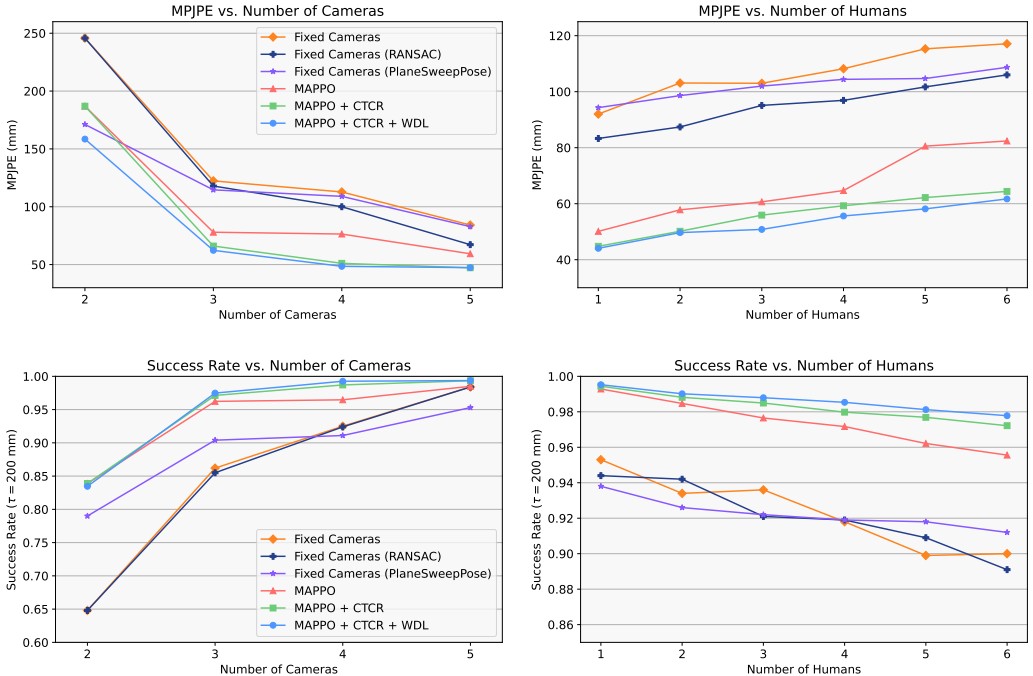

Figure 5: Comparative evaluation of mean per-joint position error (MPJPE) and success rate ($\tau = 200$mm) between the baseline methods and the proposed approaches. **Left column:** average performance w.r.t. the number of camera agents employed in environments containing one to six humans. **Right column:** average performance w.r.t. the number of humans in the environment, where three to five cameras are utilized.

## 4.2 EVALUATION METRICS

We use Mean Per Joint Position Error (MPJPE) as our primary evaluation metric, which measures the difference between the ground truth and the reconstructed 3D pose on a per-frame basis. However, using MPJPE alone may not provide a complete understanding of the robustness of a multi-camera collaboration policy for two reasons: Firstly, cameras adjusting their perception quality may take multiple frames to complete, and secondly, high peaks in MPJPE may be missed by the mean aggregation. To address this, we introduce the "success rate" metric, which evaluates the smooth execution and robustness of the learned policies. Success rate is calculated as the ratio of frames in an episode with MPJPE lower than $\tau$. Formally, $\text{SuccessRate}(\tau) = P(\text{MPJPE} \leq \tau)$. This metric is a temporal measure that reflects the integrity of multi-view coordination. Poor coordination may cause partial occlusions or too many overlapping perceptions, leading to a significant increase in MPJPE and a subsequent decrease in the success rate.

## 4.3 RESULTS AND ANALYSIS

The learning-based control policies were trained in a total of 28 instances of the BlankEnv, where each instance uniformly contained 1 to 6 humans. Each training run consisted of 700,000 steps, which corresponds to 1,000 training iterations. To ensure a fair evaluation, we report the mean metrics based on the data from the latest 100 episodes, each comprising 500 steps. The experiments were conducted in a $10\text{m} \times 10\text{m}$ area, where the cameras and humans interacted with each other.

**Active vs. Passive** To show the necessity of proactive camera control, we compare the active camera control methods with three passive methods, *i.e.*, Fixed Camera, Fixed Camera (RANSAC), and Fixed Camera (PlaneSweepPose). "Fixed Cameras" denotes that the poses of the cameras are fixed, hanging $3m$ above ground and $-35°$ camera pitch angles. The placements of these fixed cameras are carefully determined with strong priors, *e.g.*, right-angle, triangle, square, and pentagon formations for 2, 3, 4, and 5 cameras, respectively. "RANSAC" denotes the method that uses RANSAC (Fischler & Bolles, 1981) for enhanced triangulation. "PlaneSweepPose" represents the off-the-shelf learning-based method (Lin & Lee, 2021b) for multi-view 3D HPE. Please refer to Appendix E.2 for more implementation details. We show the MPJPE and Success Rate versus a

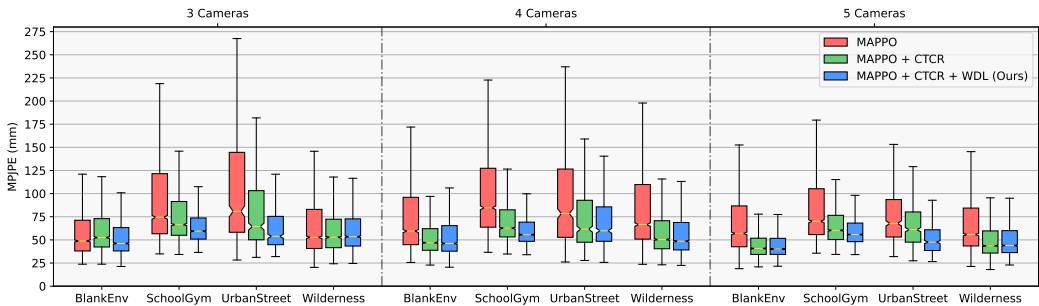

Figure 6: Evaluation results for testing our multi-camera collaboration policy (three to five cameras) in four different environments that have six walking humans (including the target human).

different number of cameras in Fig. 5. We observe that all passive baselines are being outperformed by the active approaches due to their inability to adjust camera views against dynamic occlusions. The improvement of active approaches is especially significant when the number of cameras is less, i.e. when the camera system has little or no redundancy against occlusions. Notably, the MPJPE attained by our 3-camera policy is even lower than the MPJPE from 5 Fixed cameras. This suggests that proactive strategies can help reduce the number of cameras necessary for deployment.

**The Effectiveness of CTCR and WDL**    We also perform ablation studies on the two proposed modules (CTCR and WDL) to analyze their benefits to performance. We take "MAPPO" as the active-camera baseline for comparison, which is our method but trained instead by a shared global reconstruction reward and without world dynamics modelling. Fig. 5 shows a consistent performance gap between the "MAPPO" baseline and our methods (MAPPO + CTCR + WDL). The proposed CTCR mitigates the credit assignment issue by computing the weighted marginal contribution of each camera. Also, CTCR promotes faster convergence. Training curves are shown in Appendix Fig. 13. Training with WDL objectives further improves the MPJPE metric for our 2-Camera model. However, its supporting effect gradually weakens with the increasing number of cameras. We argue that is caused by the more complex dynamics involved with more cameras simultaneously interacting in the same environment. Notably, we observe that the agents trained with WDL are of better generalization in unseen scenes, as shown in Fig. 6.

**Versus Other Active Methods**    To show the effectiveness of the learned policies, we further compare our method with other active multi-camera formation control methods in 3-cameras BlankEnv. "MAPPO" (Yu et al., 2021) and AirCapRL (Tallamraju et al., 2020) are two learning-based methods based on PPO (Schulman et al., 2017). The main difference between these two methods is the reward shaping technique, *i.e.*, AirCapRL additionally employs multiple carefully-designed rewards (Tallamraju et al., 2020) for learning. We also programmed a rule-based fixed formation control method (keeping an equilateral triangle, spread by $120°$) to track the target person. Results are shown in Table 1. Interestingly, these three baselines achieve comparable performance. Our method outperforms them, indicating a more effective multi-camera collaboration strategy for 3D HPE. For example, our method learns a spatially spread-out formation while automatically adjusting to avoid impending occlusion.

**Generalize to Various Scenarios**    We train the control policies in BlankEnv while testing them in three other realistic environments (SchoolGym, UrbanStreet, and Wilderness) to evaluate their generalizability to unseen scenarios. Fig. 6 shows that our method consistently outperforms baseline

Table 1: A comparison with other active control methods on the control of three cameras. We also report fixed camera results as a reference. MPJPE is averaged over six environment instances, each with a different number of humans (1 to 6, respectively).

|  | Fixed Cameras | MAPPO | AirCapRL (Reproduced) | Rule-based Formation | **Ours** |
|---|---|---|---|---|---|
| MPJPE (mm)↓ | 122.4 | 71.6 | 70.2 | 70.0 | **56.7** |

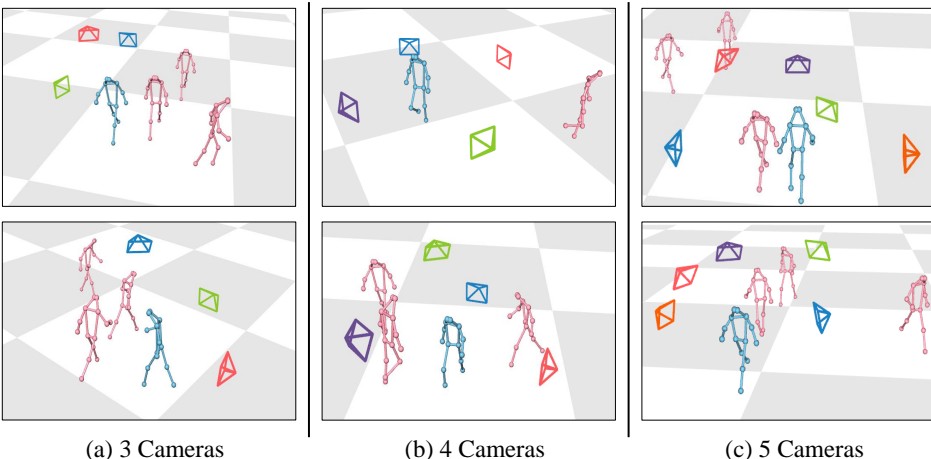

|     (a) 3 Cameras     |     (b) 4 Cameras     |     (c) 5 Cameras     |

Figure 7: Visualizations for the emergent formations in controlling three to five cameras. The target person is colored in blue and others are colored in pink. The camera agents learn to spread out and avoid occlusions while maintaining good reconstruction accuracy without a centralized controller.

methods with lower variance in MPJPE during the evaluations in three test environments. We report the results in the BlankEnv as a reference.

**Qualitative Analysis** In Figure 7, we show six examples of the emergent formations of cameras under the trained policies using the proposed methods (CTCR + WDL). The camera agents learn to spread out and ascent above humans to avoid occlusions and collision. Their placements in an emergent formation are not assigned by other entities but rather determined by the decentralized control policies themselves based on local observations and agent-to-agent communication. For more vivid examples of emergent formations, please refer to the project website for the demo videos.[1] For more analysis on the behaviour mode of our 3-, 4- and 5-camera models, please refer to Appendix Section H.

## 5 Conclusion and Discussion

To our knowledge, this paper presents the first proactive multi-camera system targeting the 3D reconstruction in a dynamic crowd. It is also the first study regarding proactive 3D HPE that promptly experimented with multi-camera collaborations at different scales and scenarios. We propose CTCR to alleviate the multi-agent credit assignment issue when the camera team scales up. We identified multiple auxiliary tasks that improve the representation learning of complex dynamics. As a final note, we release our virtual environments and the visualization tool to facilitate future research.

**Limitations and Future Directions** Admittedly, a couple of aspects of this work have room for improvement. Firstly, the camera agents received the positions for their teammates via non-disrupting broadcasts of information, which may be prone to packet losses during deployment. One idea is to incorporate a specialized protocol for multi-agent communication into our pipeline, such as ToM2C(Wang et al., 2022). Secondly, intended initially to reduce the communication bandwidth (for example, to eliminate the need for image transmission between cameras), the current pipeline comprises a Human Re-Identification module requiring a pre-scan memory on all to-be-appeared human subjects. For some out-of-distribution humans' appearances, the current ReID module may not be able to recognize. Though ACTIVE3DPOSE can accommodate a more sophisticated ReID module (Deng et al., 2018) to resolve this shortcoming. Thirdly, the camera control policy requires accurate camera pose, which may need to develop a robust SLAM system (Schmuck & Chli, 2017; Zhong et al., 2018b) to work in dynamic environments with multiple cameras. Fourthly, the motion patterns of targets in virtual environments are based on manually designed animations, which will lead to the poor generalization of the agents on unseen motion patterns. In the future, we can enrich the diversity by incorporating a cooperative-competitive multi-agent game (Zhong et al., 2021) in training. Lastly, we assume near-perfect calibrations for a group of mobile cameras, which might be complicated to sustain in practice. Fortunately, we are seeing rising interest in parameter-free pose estimation (Gordon et al., 2022; Ci et al., 2022), which does not require online camera calibration and may help to resolve this limitation.

---

[1]Project Website for demo videos: https://sites.google.com/view/active3dpose

## 6 ETHICS STATEMENT

Our research into active 3D HPE technologies has the potential to bring many benefits, such as biomechanical analysis in sports and automated video-assisted coaching (AVAC). However, we recognize that these technologies can also be misused for repressive surveillance, leading to privacy infringements and human rights violations. We firmly condemn such malicious acts and advocate for the fair and responsible use of our virtual environment, UNREALPOSE, and all other 3D HPE technologies.

## ACKNOWLEDGEMENT

The authors would like to thank Yuanfei Wang for discussions on world models; Tingyun Yan for his technical support on the first prototype of UNREALPOSE. This research was supported by MOST-2022ZD0114900, NSFC-62061136001, China National Post-doctoral Program for Innovative Talents (Grant No. BX2021008) and Qualcomm University Research Grant.

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

## A TRAINING ALGORITHM PSUEDOCODE

---

**Algorithm 1** Learning Multi-Camera Collaboration (CTCR + WDL)

---

1: **Initialize:** $n$ agents with a tied-weights MAPPO policy $\pi$ and mixture density models (MDN) for WDL prediction models $\{(P_{\text{self}}, P_{\text{reward}}, P_{\text{peer}}, P_{\text{tgt}}, P_{\text{pd}})\}_\pi$, $\mathcal{E}$ parallel environment rollouts
2: **for** Iteration $= 1, 2, \ldots M$ **do**
3:     In each $\mathcal{E}$ environment rollouts, each agent $i \in [\![n]\!]$ collects trajectory with length $T$:
$$\boldsymbol{\tau} = \left[(\tilde{o}_i^t, \tilde{o}_i^{t+1}, a_i^t, r_i^t, h_i^{t-1}, s^t, \boldsymbol{a}^{t-1}, r^t)\right]_{t=1}^T$$
4:     Substitute individual reward of each agent $r^t$ with CTCR: $r_i^t \leftarrow \text{CTCR}(i)$ (equation 6)
5:     For each step in $\boldsymbol{\tau}$, compute advantage estimates $\hat{A}_i^1, \ldots, \hat{A}_i^T$ with GAE (Schulman et al., 2015) for each agent $i \in [\![n]\!]$
6:     Yield a training batch $\mathcal{D}$ of the size $\mathcal{E} \times T \times n$
7:     **for** Mini-batch SGD Epoch $= 1, 2, \ldots, K$ **do**
8:         Sample a stochastic mini-batch of size $B$ form $\mathcal{D}$, which $B = |\mathcal{D}|/K$
9:         Compute $z_i^t$ and $h_i^t$ using the encoder model $E_\pi$
10:         Compute PPO-CLIP objective loss $\mathcal{L}_{\text{PPO}}$, global critic value loss $\mathcal{L}_{\text{Value}}$, adaptive KL loss $\mathcal{L}_{\text{KL}}$
11:         Compute the objectives that constitutes $\mathcal{L}_{\text{WDL}}$:
- Self-State Prediction Loss    $\mathcal{L}_{\text{self}} = -\mathbb{E}_{\boldsymbol{\tau}}[\log P(\xi_i^{t+1}|z_i^t, h_i^t, a_i^t)]$
- Reward Prediction Loss    $\mathcal{L}_{\text{reward}} = -\mathbb{E}_{\boldsymbol{\tau}}[\log P(r^t|z_i^t, h_i^t, a_i^t)]$
- Peer-State Prediction Loss    $\mathcal{L}_{\text{peer}} = -\mathbb{E}_{\boldsymbol{\tau}}[\log P(\xi_{-\boldsymbol{i}}^{t+1}|z_i^t, h_i^t, a_i^t)]$
- Target Prediction Loss    $\mathcal{L}_{\text{tgt}} = -\mathbb{E}_{\boldsymbol{\tau}}[\log P(p_{\text{tgt}}^{t+1}|z_i^t, h_i^t, p_{\text{tgt}}^t)]$
- Pedestrians Prediction Loss    $\mathcal{L}_{\text{pd}} = -\mathbb{E}_{\boldsymbol{\tau}}[\log P(p_{\text{pd}}^{t+1}|z_i^t, h_i^t, p_{\text{pd}}^t)]$
12:         $\mathcal{L}_{\text{Train}} = \lambda_{\text{PPO}}\mathcal{L}_{\text{PPO}} + \lambda_{\text{Value}}\mathcal{L}_{\text{Value}} + \beta_{\text{KL}}\mathcal{L}_{\text{KL}} + \lambda_{\text{WDL}}\mathcal{L}_{\text{WDL}}$
13:         Optimize $\mathcal{L}_{\text{Train}}$ w.r.t to the current policy parameter $\theta_\pi$

---

## B    UNREALPOSE: ACCESSORIES AND MISCELLANEOUS ITEMS

Our UnrealPose virtual environment supports different active vision tasks, such as active human pose estimation and active tracking. This environment also supports various settings ranging from single-target single-camera settings to multi-target multi-camera settings. Here we provide a more detailed description of the three key characteristics of UnrealPose:

**Realistic** The built-in navigation system governs the collision avoidance movements of virtual humans against dynamic obstacles. In the meantime, it also ensures diverse generations of walking trajectories. These features enable the users to simulate a realistic-looking crowd exhibiting socially acceptable behaviors. We have also provided several pre-set scenes, *e.g.*, school gym, wilderness, urban crossing, and *etc*. These scenes have notable differences in illuminations, terrains, and crowd appearances to reflect the dramatically different-looking scenarios in real-life.

**Extensive Configuration** The environment can be configured with different numbers of humans and cameras and swapped across other scenarios with ease, which we demonstrated in Fig 4(a-d). Besides simulating walking human crowds, the environment incorporates over 100 Mocap action sequences with smooth animation interpolations to enrich the data variety for other MoCap tasks.

**RL-Ready** We use the UnrealCV (Qiu et al., 2017) plugin as the medium to acquire images and annotations from the environment. The original UnrealCV plugin suffers from unstable data transfer and unexpected disconnections under high CPU workloads. To ensure fast and reliable data acquisition for large-scale MARL experiments, we overhauled the communication module in the UnrealCV plugin with inter-process communication (IPC) mechanism, which eliminates the aforementioned instabilities.

### B.1    VISUALIZATION TOOL

We provide a visualization tool to facilitate per-frame analysis of the learned policy and reconstruction results (shown in Fig. 8). The main interface consists of four parts: (1) Live 2D views from all cameras. (2) 3D spatial view of camera positions and reconstructions. (3) Plot of statistics. (4) Frame control bar. This visualization tool supports different numbers of humans and cameras. Meanwhile, it is written in Python to support easy customization.

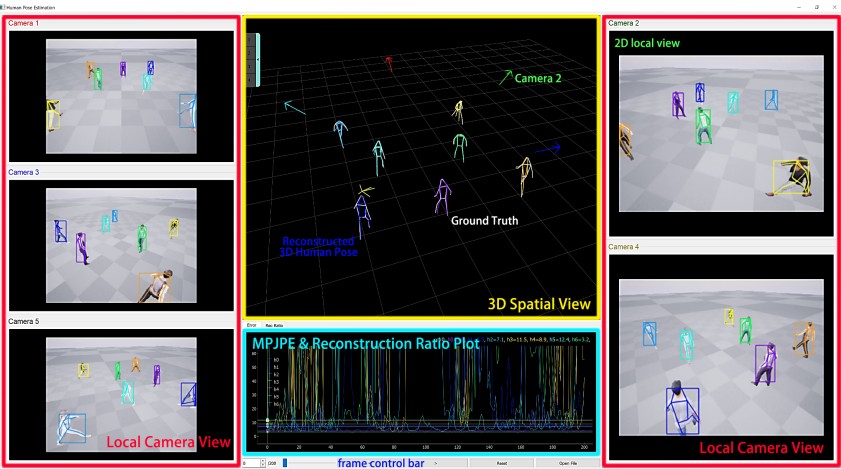

Figure 8: Visualization Tool. RGB videos at the two sides show the local camera view for each individual camera. The center part shows the live 3D reconstructions and corresponding MPJPE.

### B.2    LICENSE

All assets used in the environment are commercially-available and obtained from the UE4 Marketplace. The environment and tools developed in this work are licensed under Apache License 2.0.

## C  OBSERVATION PROCESSING

Fig.9 shows the pipeline of the observation processing. Each camera observes an RGB image and detects the 2D human poses and IDs via the Perception Module described in the main paper. The camera pose, 2D human poses and IDs are then broadcast to other cameras for multi-view 3D triangulation. The human position is calculated as the median of the reconstructed joints. Human orientation is calculated from the cross product of the reconstructed shoulder and spine.

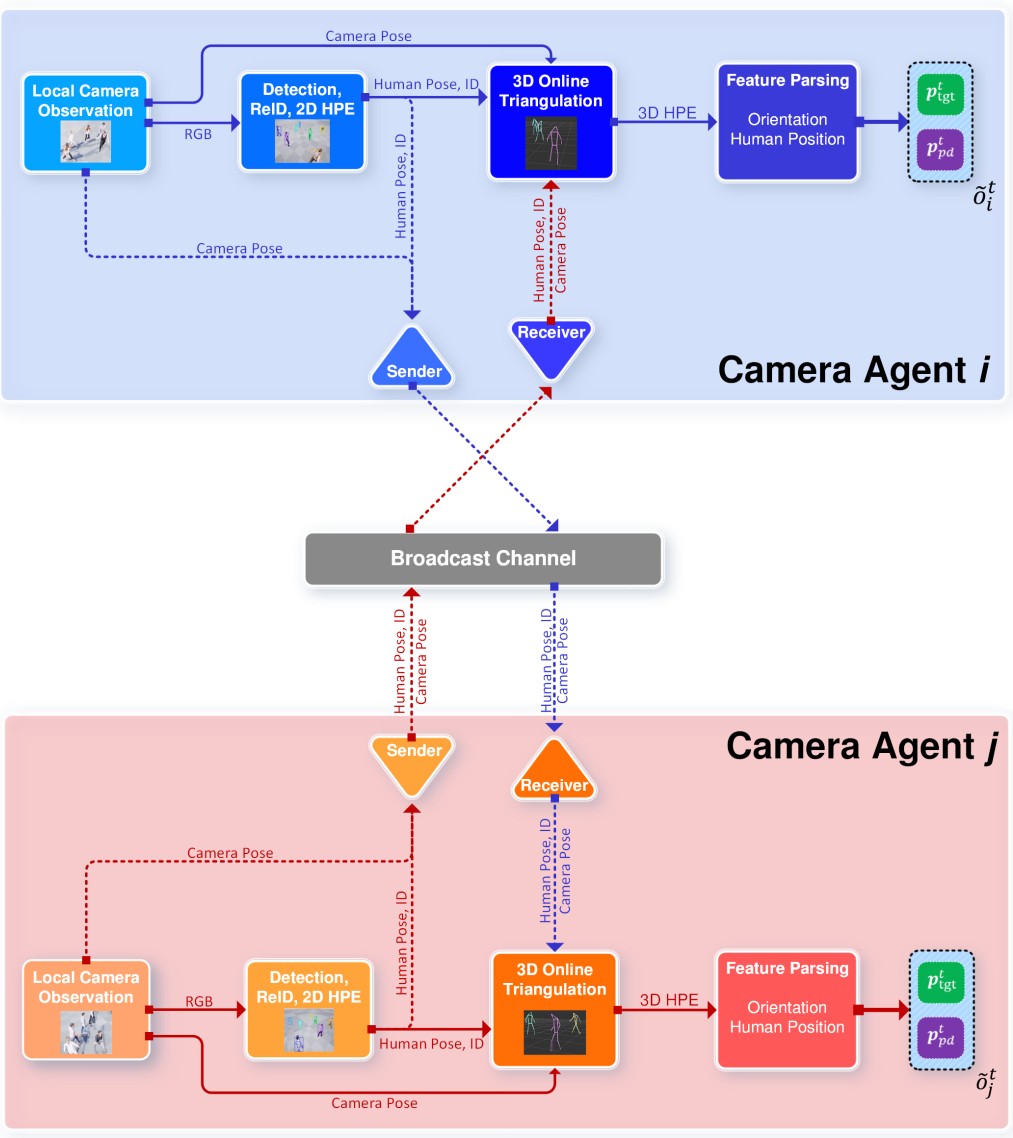

Figure 9: Decentralized Observation Processing

## D   IMPLEMENTATION DETAILS

### D.1   TRAINING DETAILS

All control policies are trained in the `BlackEnv` scene. At the testing stage, we apply zero-shot transfer for the learned policies to three realistic scenes: `SchoolGym`, `UrbanStreet`, and `Wilderness`.

To simulate a dynamic human crowd that performs high random behaviors, we sample arbitrary goals for each human and employ the built-in navigation system to generate collision-free trajectories. Each human walks at a random speed. To ensure the generalization across a different number of humans, we train our RL policy with a mixture of environments of 1 to 6 humans. The learning rate is set to $5 \times 10^{-4}$ with scheduled decay during the training phase. The annealing schedule for the learning rate is detailed in Table. 2. The maximum episode length is 500 steps, discounted factor $\gamma$ is 0.99 and the GAE horizon is 25 steps. Each sampling iteration produces a training batch with the size of 700 steps, then we perform 16 iterations on every training batch with 2 SGD mini-batch updates for each iteration (*i.e.* SGD batch size = 350).

Table 2 shows the common training hyper-parameters shared between the baseline models (MAPPO) and all of our methods. Table. 4 shows the hyperparameters for the WDL module.

Table 2: Common hyperparameters for models in baselines and our methods

| Common Hyperparameter | Baseline, Our Methods |
|---|---|
| Max Episode Length | 500 |
| Rollout Fragment Length | 25 |
| Number of Rollouts | 28 |
| Train Batch Size | 700 |
| SGD Mini-batch Size | 350 |
| Number of SGD Iterations (Sample Reuse) | 16 |
| gamma ($\gamma$) | 0.99 |
| GAE lambda ($\lambda$) | 1.0 |
| Initial KL Coefficient | 0.2 |
| $KL_{Target}$ | 0.01 |
| Entropy Regulation Coefficient | 0.0 |
| Value Function Loss Coefficient | 0.1 |
| Value Function Clipping | 1000 |
| Gradient Clipping | 50 |
| Initial Learning Rate | 0.0005 |
| Learning Rate Schedule (lr, end step) | [(0, 5E-4), (200E3, 5E-4), (200E3, 1E-4), (400E3, 1E-4), (600E3, 5E-5), (600E3, 5E-5)] |
| LSTM Cell Size | 128 |
| Encoder Hidden Layers | [128, 128, 128] |
| Value Network Hidden Layers | [128] |
| Actor Network Hidden Layers | [128] |

### D.2   DIMENSIONS OF FEATURE TENSORS IN THE CONTROLLER MODULE

Table 3 serves as a complementary description for Eqn. 1, 2, 3, 4. The table shows the dimensions of the feature tensors used in the controller module. "B" denotes the batch size. In the current model design, the dimension of local observation is adjusted based on the maximum number of camera agents ($N\_cam_{max}$) and the maximum number of observable humans ($N\_human_{max}$) in an environment. In our experiments, $N\_human_{max}$ has been set to 7. The observation pre-processor will zero-pad each observation to a length equal $N\_human_{max} \times 18$ if the current environment

instance has less than $N\_human_{\mathrm{max}}$ humans. "9" and "18" correspond to the feature length for a camera and a human, respectively. The MDN of the Target Prediction module has 16 Gaussian components, in which each component outputs $(\phi, \mu_x, \sigma_x, \mu_y, \sigma_y)$ and $\phi$ is the weight parameter of a component. The current implementation of MDN only predicts the x-y location of a human, which is a simplification since the z coordinate of a simulated human barely changes across an episode compared to the x and y coordinates. The dimension of an MDN output has a length of 80 and the exact prediction is produced by $\phi$-weighted averaging. In Eqn. 4, $\{(\phi, \mu, \sigma)\}_{\mathrm{MDN_{tgt}}}$ is an encoded feature produced by passing the MDN output to a 2-layer MLP that has an output dimension of 128.

Table 3: Dimensions of the feature tensors used in the controller module

| Feature | Shape |
|---|---|
| $\hat{o}_i$ | (B, $N\_cam_{\mathrm{max}}{\times}9{+}N\_human_{\mathrm{max}}{\times}18$) |
| $z_i$ | (B, 128) |
| $h_i$ | (B, 128) |
| $\hat{p}_{\mathrm{tgt}}, \hat{p}_{\mathrm{pd}}$ | (B, 128) |
| $\{(\phi, \mu, \sigma)\}_{\mathrm{MDN_{tgt}}}$ | (B, 128) |
| $e_i$ | (B, 384) |

### D.3 COMPUTATIONAL RESOURCES

We used 15 Ray workers for each experiment to ensure consistency in the training procedure. Each worker carries a Gym vectorized environment consisting of 4 actual environment instances. Each worker demands approximately $3.7$ GB of VRAM. We run each experiment with 8 NVIDIA RTX 2080 Ti GPUs. Depending on the number of camera agents, the total training hours required for an experiment to run 500k steps will vary between 4 to 12 hours.

Table 4: Hyperparamters for WDL

| WDL Hyperparameter | Our Methods |
|---|---|
| Prediction Loss Coefficient ($\lambda_{\mathrm{WDL}}$) | 1.0 |
| Camera Prediction Coefficient | 1.0 |
| Other Camera Prediction Coefficient | 1.0 |
| Reward Prediction Coefficient | 1.0 |
| Human Prediction Coefficient | 1.0 |
| Pedestrian Prediction Coefficient | 0.1 |
| MDN: Hidden Layers | [128, 128] |
| MDN: Number of Gaussian Distributions | 16 |

### D.4 TRAINING CURVE

Fig. 13 shows the training curves of our method and the baseline method. We can find that our methods converge faster and improve reconstruction accuracy than the baseline method.

### D.5 TOTAL INFERENCE TIME

Our solution can run in real-time. Table 5 reports the inference time for each module of the proposed Active3DPose pipeline.

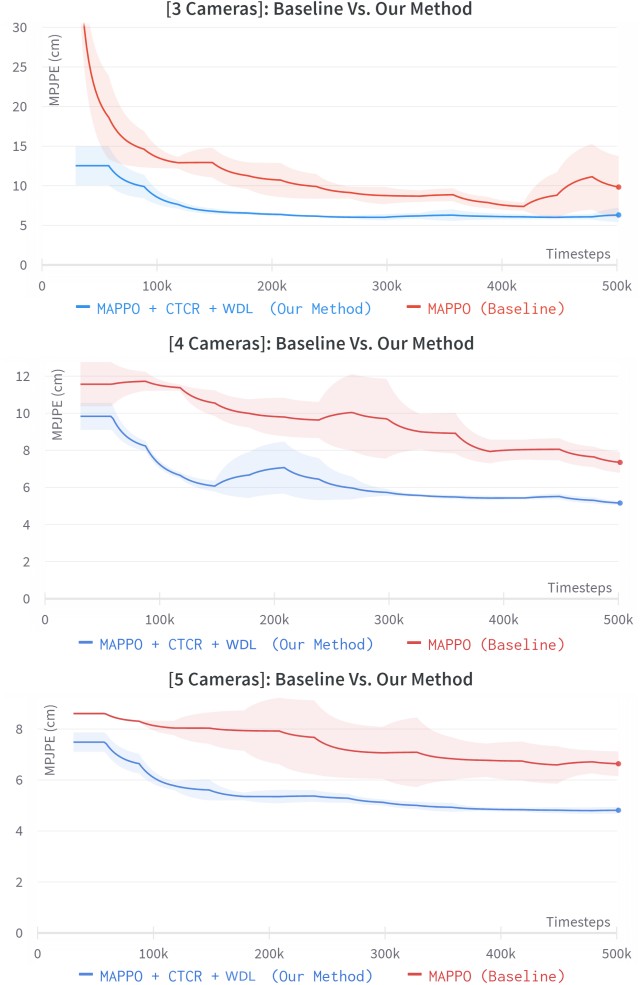

Figure 10: Comparing the training curves (MPJPE (cm) vs. Environment Steps) for **3, 4, 5** cameras policies trained to 500k timesteps for the baseline and our method. Intervals for one standard deviation are shown, and each interval is computed based on samples from 3 different runs on different seeds.

Table 5: The total inference speed of our current implementation is around 8 FPS. The average time data are generated on a computing node with one NVIDIA RTX 2080TI GPU and one Intel Xeon E5-2699A CPU. These data are from testing our 5-camera model over 2500 environment steps in the Blank Environment.

|  | **Average Time (ms)** |
|---|---|
| Human Detection | 20.05 |
| ReID Module | 28.88 |
| 2D Pose Estimation | 31.16 |
| 3D Triangulation | 1.43 |
| Observation Processing | 3.97 |
| RL Model Inference | 40.76 |
| **Total Inference Time** | 126.25 |

# E ADDITIONAL EXPERIMENT RESULTS

## E.1 ABLATION STUDY ON WDL OBJECTIVES

We perform a detailed ablation study regarding the effect of each WDL sub-task on the model performance. As shown in Fig. 11, we can observe that the MPJPE metric gradually decreases as we incorporate more WDL losses. This aligns with our assumptions that training the model with world dynamics learning objectives will promote the model's ability to capture a better representation of future states, which in turn increases performance. Our method additionally demonstrates the importance of incorporating information regarding the target's future state into the encoder's output features. Predicting the target's future states should not only be used as an auxiliary task but should also directly influence the inference process of the actor model.

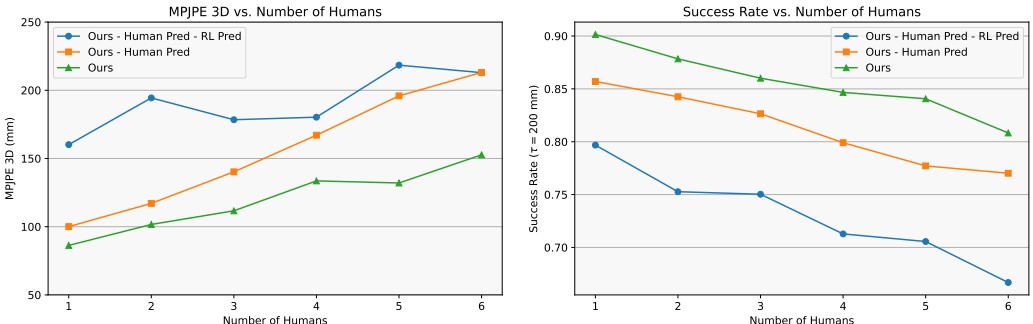

Figure 11: Detailed ablation study on WDL modules in the 2-cameras environment. Here we commence the study by running three separate experiments differed by the number of auxiliary tasks involved. A minus sign **(-)** followed by the name of a module indicates the absence of this module in this trial. **RLPred**: learn agent's forward dynamics, **Human Pred**: predict future target position and predict future pedestrian position. **Ours** is our fully implemented method (MAPPO+WDL) for controlling 2 camera agents.

## E.2 BASELINE — FIXED-CAMERAS

In addition to the triangulation and RANSAC baselines introduced in the main text, we compare and elaborate on two more baselines that use fixed cameras: (1) fixed cameras with RANSAC-based triangulation and temporal smoothing (TS) (2) an off-shelf 3D pose estimator PlaneSweepPose (Lin & Lee, 2021a).

In the temporal smoothing baseline, we applied a low-pass filter (Casiez et al., 2012) and temporal fusion where the algorithm will fill in any missing key points in the current frame with the detected key points from the last frame.

In the PlaneSweep baseline, as per the official instructions, we train three separate models (3 cams to 5 cams) with the same camera setup as in our testing scenarios. We have tested the trained models in different scenarios and reported the MPJPE results in Table 6, 7, 8 and 9. Note that, this off-shelf pose estimator performs better than the Fixed-Camera Baseline (Triangulation) but still underperforms compared to our active method. Fig. 12 illustrates the formations of the fixed camera baselines.

Table 6: Four fixed-camera baselines compared to our method in the 6-human Blank environment. Results for all baselines are evaluated over 20 episodes of 500 steps. **RANSAC:** RANSAC-based Triangulation **TS:** Temporal Smoothing with temporal fusion **PlaneSweep:** PlaneSweepPose (Lin & Lee, 2021a).

| BlankEnv | Fixed Cameras (Triangulation) | Fixed Cameras (RANSAC) | Fixed Cameras (RANSAC+TS) | Fixed Cameras (PlaneSweep) | Ours |
|---|---|---|---|---|---|
| 3 Cameras | 142.0 | 129.0 | 123.5 | 123.3 | **73.3** |
| 4 Cameras | 122.3 | 116.6 | 106.0 | 116.4 | **54.7** |
| 5 Cameras | 87.0 | 72.3 | 77.7 | 86.5 | **51.6** |

Table 7: Four fixed-camera baselines compared to our method in the 6-human SchoolGym environment. Results for all baselines are evaluated over 20 episodes of 500 steps. **RANSAC:** RANSAC-based Triangulation **TS:** Temporal Smoothing with temporal fusion **PlaneSweep:** PlaneSweep-Pose (Lin & Lee, 2021a).

| SchoolGym | Fixed Cameras (Triangulation) | Fixed Cameras (RANSAC) | Fixed Cameras (RANSAC+TS) | Fixed Cameras (PlaneSweep) | Ours |
|---|---|---|---|---|---|
| 3 Cameras | 149.1 | 149.8 | 144.1 | 132.6 | **99.4** |
| 4 Cameras | 133.3 | 124.5 | 124.7 | 120.3 | **67.7** |
| 5 Cameras | 100.2 | 76.8 | 83.0 | 92.8 | **64.6** |

Table 8: Four fixed-camera baselines compared to our method in the 6-human UrbanStreet environment. Results for all baselines are evaluated over 5 episodes of 500 steps. **RANSAC:** RANSAC-based Triangulation **TS:** Temporal Smoothing with temporal fusion **PlaneSweep:** PlaneSweepPose (Lin & Lee, 2021a).

| UrbanStreet | Fixed Cameras (Triangulation) | Fixed Cameras (RANSAC) | Fixed Cameras (RANSAC+TS) | Fixed Cameras (PlaneSweep) | Ours |
|---|---|---|---|---|---|
| 3 Cameras | 513.4 | 210.1 | 194.2 | 160.3 | **103.6** |
| 4 Cameras | 279.4 | 190.8 | 174.5 | 154.5 | **66.4** |
| 5 Cameras | 136.7 | 94.1 | 94.4 | 106.5 | **61.9** |

Table 9: Four fixed-camera baselines compared to our method in the 6-human Wilderness environment. Results for all baselines are evaluated over 20 episodes of 500 steps. **RANSAC:** RANSAC-based Triangulation **TS:** Temporal Smoothing with temporal fusion **PlaneSweep:** PlaneSweepPose (Lin & Lee, 2021a).

| Wilderness | Fixed Cameras (Triangulation) | Fixed Cameras (RANSAC) | Fixed Cameras (RANSAC+TS) | Fixed Cameras (PlaneSweep) | Ours |
|---|---|---|---|---|---|
| 3 Cameras | 234.0 | 159.6 | 160.5 | 139.2 | **72.6** |
| 4 Cameras | 165.6 | 143.9 | 146.6 | 129.5 | **71.3** |
| 5 Cameras | 108.1 | 74.9 | 86.5 | 95.1 | **54.3** |

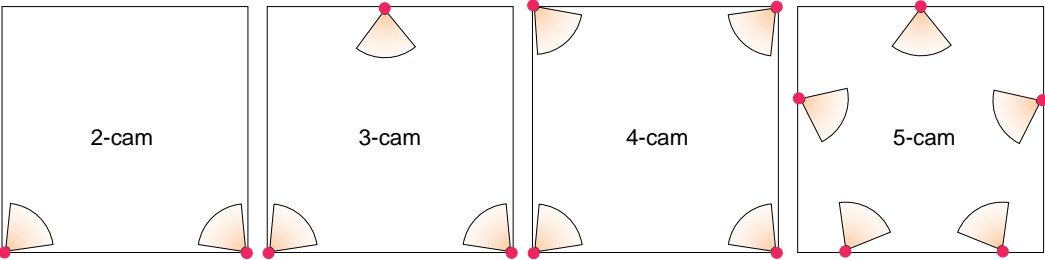

Figure 12: Illustration of fixed camera placements

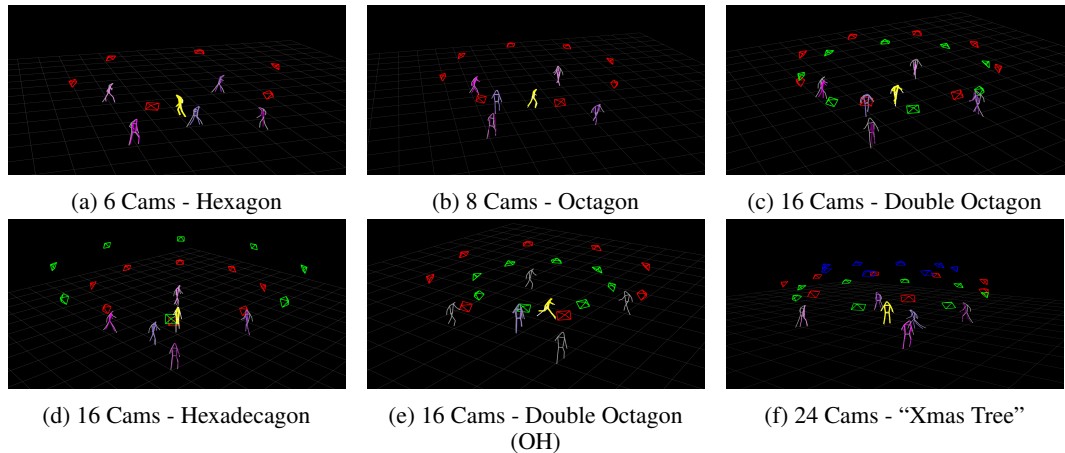

(a) 6 Cams - Hexagon

(b) 8 Cams - Octagon

(c) 16 Cams - Double Octagon

(d) 16 Cams - Hexadecagon

(e) 16 Cams - Double Octagon (OH)

(f) 24 Cams - "Xmas Tree"

Figure 13: In search of a topology of a fixed-camera system that could outperform our 5-camera model, we have designed six more 3D topologies that contain 8 to 24 cameras.

Camera placements for all fixed-camera baselines are shown in Fig. 12. These formations are carefully designed not to disadvantage the fixed-camera baselines on purpose. Especially for the 5-cameras pentagon formation, which helps the fixed-camera baseline to obtain satisfactory performance on the 5-camera setting as shown in Tables 6, 7, 8 and 9.

Table 10: Comparing six more fixed-camera network topologies with our 5-camera method. Although these topologies have significantly more cameras than our 5-active-camera model, the fixed-camera approach cannot close the performance gap between fixed-camera systems and the learned policy. We also observe inversions in MPJPE when increasing the number of cameras from 8 to 16 to 24. These results suggest that the fixed-camera system has saturated the reconstruction accuracy beyond 8 cameras, and there are no further upsides to increasing the scale. We reason this is due to the errors and biases in the 2D pose model that subsequently affect the triangulation algorithm's effectiveness.

|  | MPJPE (mm) |
| --- | --- |
| **6 Cams - Hexagon** | 88.0 |
| **8 Cams - Octagon** | 78.5 |
| **16 Cams - Double Octagon** | 86.7 |
| **16 Cams - Hexadecagon** | 81.7 |
| **16 Cams - Double Octagon (OH)** | 94.9 |
| **24 Cams - "Xmas Tree"** | 87.0 |
| **Ours, 5 Cams, active cameras** | 51.6 |

### E.3 Our Method Enhanced with RANSAC-based Triangulation

RANSAC is a generic technique that can be used to improve triangulation performance. In this experiment, we also train and test our model with RANSAC. The final result (Table 11) shows a further improvement on our original triangulation version.

Table 11: Evaluate the effect of adding RANSAC-based Triangulation to our method. Both models are trained from scratch on a mix of Blank environment instances with 1 to 6 humans. MPJPE(mm) is averaged over the training results in six different configurations of Blank environment instances (1-6 humans).

|  | Ours (5 Cams) | Ours (5 Cams + RANSAC) |
|---|---|---|
| MPJPE (mm)↓ | 46.7 | **35.0** |

## F Generating Safe and Smooth Trajectories

### F.1 Collision Avoidance

In order to generate safe trajectories, in this section, we introduce and evaluate two different ways to enforce collision avoidance between cameras and humans.

**Obstacle Collision Avoidance (OCA)**  OCA resembles a feed-forward PID controller on the final control outputs before execution. Concretely, OCA adds a constant reverse quantity to the control output if it detects any surrounding objects within its safety range. This "detouring" mechanism safeguards the cameras from possible collisions and prevents them from making dangerous maneuvers.

**Action-Masking (AM)**  AM also resembles a feed-forward controller but is instead embedded into the forward step of the deep-learning model. At each step, AM module first identifies the dangerous actions among all possible actions, then modifies the probabilities (output by the policy model) of choosing the hazardous actions to be zero so that the learning-based control policy will never pick them. Note that AM must be trained with the MARL policy model.

We proposed the minimum Camera-Human distance (in which the scope includes the target and the pedestrians) as the safety metric. It measures the distance between the closest human and the camera at a timestep. Fig. 14 shows the histograms of Min Camera-Human distance sampled over five episodes of 500 steps.

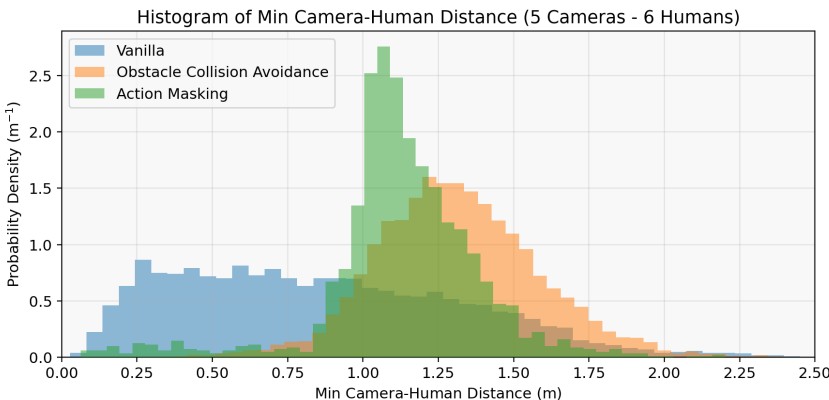

Figure 14: Histograms of minimum Camera-to-Human distances to demonstrate the safety integrity level for three different **5-Cameras** methods. **Vanilla**: original deep MARL policy without any safety feature. **Obstacle Collision Avoidance**: deep MARL policy added with OCA module **Action Masking**: deep MARL policy trained with AM module.

## F.2 TRAJECTORY SMOOTHING

**Exponential Moving Average (EMA)**   To generate smooth trajectories, we introduce EMA to smooth the outputs of our learned policy model. EMA is a common technique used in smoothing time-series data. In our case, the smoothing operator is defined as :

$$\hat{a}_t = \hat{a}_{t-1} + \eta \cdot (a_t - \hat{a}_{t-1})$$

Where $a_t$ is the action (or control signal) output by the model at the current step, $\hat{a}_{t-1}$ is the smoothed action from the last step and $\eta$ is the smoothing factor. A smaller $\eta$ results in greater smoothness. $\hat{a}_t$ is the smoothed action that the camera will execute.

## F.3 ROBUSTNESS OF THE LEARNED POLICY

In this section, we evaluate the robustness of our model on different external perturbations to the control signal. In conclusion, our model shows resilience against the delay effect and random noise.

**Delay**   EMA also brings a delay effect while smoothing the generated trajectory. The level of smoothing positively correlates with a larger delay factor. Here, we evaluate our model's robustness to the EMA simulated delay.

**Random Action Noise**   Control devices in real-life are inevitably affected by errors. For example, the control signal of a controller may be over-damped or may overshoot. We simulated this type of random error by multiplying the output action by a uniformly-sampled noise.

In Table 12, we intend to observe the effects of EMA delay and random action noise on the reconstruction accuracy of our model, marked as "Vanilla".

Table 12: Adding delay (EMA) and action noise to our proposed method. In this experiment, we set $\eta = 0.50$ and noise factor between $[0.80, 1.20)$. Results are reported on 5 episodes of 500 steps.

|  | Ours(Vanilla) | Add delay(EMA) | Add action noise | Add action noise and delay(EMA) |
|---|---|---|---|---|
| **MPJPE (mm)** | (51.6) | +0.8 | +2.1 | +3.9 |

## G    MORE EXPLANATIONS ON CTCR

Figure 3 is an example of using Eq. 6 to compute CTCR for each of the three cameras. The CTCR is incentivized by the Shapley Value.The main idea is that the overall optimality needs to also account for the optimality of every possible sub-formation. For a camera agent to receive the highest CTCR possible, its current position and view must be optimal both in terms of its current formation and any sub-formation possible.

**Note:** a group of collaborating players is often referred to as a "coalition" in other literatures. Here we apply the same concept but to a group of cameras, so we used the more initiative term "formation" instead.

Eq. 6 can be further breakdown as follows:

$$\varphi_r(i) = \sum_{S \subseteq [\![n]\!] \setminus \{i\}} \frac{|S|!(n - |S| - 1)!}{n!} [r(S \cup \{i\}) - r(S)]$$

$$= \frac{1}{n} \sum_{S \subseteq [\![n]\!] \setminus \{i\}} \frac{|S|!(n - 1 - |S|)!}{(n-1)!} [r(S \cup \{i\}) - r(S)]$$

$$= \frac{1}{n} \sum_{S \subseteq [\![n]\!] \setminus \{i\}} \binom{n-1}{|S|}^{-1} [r(S \cup \{i\}) - r(S)]$$

where $[\![n]\!] = 1, 2, \ldots, n$ denotes the set of all cameras and the binomial coefficient $\binom{k}{n} = \frac{n!}{k!(n-k)!}, 0 \leq k \leq n$. $S$ denotes a formation (a subset) without camera $i$. $\binom{n-1}{|S|}$ is the number of combinations of subset $S$ (i.e., the binomial coefficient), which serves as a normalization term. $r(S)$ computes the reconstruction accuracy of the formation $S$. And $[r(S \cup \{i\}) - r(S)]$ computes the marginal improvement after adding the camera $i$ to sub-formation $S$. So this equation means we iterate over all possible $S$ and compute the marginal contribution of camera $i$ and average over all possible combinations of $(S, i)$.

Suppose we have a 3-cameras formation, similarly shown in Figure 3. So $n = 3$, the number of cameras. Let's name these cameras (1,2,3), and let's say we only care about Camera 1 for now. Since we are computing the average marginal contribution for Camera 1, we are looking for those formations that do not have Camera 1 because we want to see how much of an increase in performance resulted from the addition of Camera 1 to those formations. For all possible formations denoted by the set $[\![n]\!]$, four formations satisfy this condition, $S \subseteq [\![n]\!] \setminus \{i\} \longrightarrow S \in (\emptyset, \{2\}, \{3\}, \{2, 3\})$.

The binomial coefficient $\binom{n-1}{|S|}$ for a 2-camera sub-formation in a 3-cameras case is $\binom{2}{2}$, which makes sense because there exists only one unique combination that does not contain Camera 1, which is sub-formation $\{2, 3\}$. $r(\{2, 3\})$ computes the reconstruction accuracy of the formation $\{2, 3\}$ and $r(\{2, 3\} \cup \{1\})$ computes the reconstruction accuracy after adding Camera 1 to sub-formation $\{2, 3\}$. Their difference gives us the marginal contribution of Camera 1. As we sum over all subsets $S$ of $[\![n]\!]$ not containing Camera 1, and then divide by $\binom{n-1}{|S|}$ and the number of cameras $n$, we have the average marginal contribution of Camera 1 ($\varphi_r(\{1\})$) to the collaborative triangulated reconstruction. Further multiplying this term by $n$, you have the $\text{CTCR}_1$ as shown in Eq. 6.

# H ANALYSIS ON MODES OF BEHAVIORS OF THE TRAINED AGENTS

Here in Figure 15 we provide statistics and analysis on the behaviour mode of the agents controlled by our 3-, 4- and 5-camera policies, respectively. We are interested in understanding the characteristics of the emergent formations learned by our model. Hence we proposed three quantitative measures to understand the topology of the emergent formations: (1) the min-angle between the cameras' orientation and the per-frame-mean of the min-camera angle, (2) the camera's pitch angle, and (3) the camera-human distance. Hence we provide rigorous definitions as follows:

$$\text{min-camera angle}(i) = \min_{j \neq i} \langle \text{axis of camera } i, \text{axis of camera } j \rangle$$

$$\text{per-frame-mean of min-camera angle} = \frac{1}{n} \sum_{i \in [n]} \text{min-camera angle}(i)$$

In simpler terms, $\text{min-camera angle}(i)$ finds the minimum angle between camera $i$ and any other camera $j$. "per-frame-mean of min-camera angle" is the mean of $\text{min-camera angle}(i)$ for all camera $i$ in one frame. A positive camera's pitch angle means looking upward, and a negative camera's pitch angle means looking downward. The camera-human distance measures the distance between the target human and the given camera.

Regarding the distance between cameras and humans, the camera agents actively adjust their position relative to the target human. The cameras learn to keep a camera-human distance between 2m and 3m. It is neither too distant from the target human nor too close, violating the safety constraint. The camera agents surround the target at an appropriate distance to have a better resolution on the 2D views. In the meantime, as the safe-distance constraint is enforced during training, the camera agents are prohibited from aggressively minimizing the camera-human distance.

The histograms for the camera's pitch angle suggest that the cameras mostly maintain negative pitch angles. Their preferred strategy is to hover over the humans and capture images at a higher altitude. This is likely because of emergent occlusion avoidance. The cameras also emerge to fly at an even level with the humans (where the pitch angle approximately equals 0) to capture more accurate 2D poses of the target human. This propensity is apparent from the peaks at $x = 0$ angles in the histograms. A relatively wide distribution of the pitch angle histogram suggests that the camera formation is spread out in the 3D space and dynamic adjustments of flying heights and pitch angles by the camera agents.

For the average angle between the cameras' orientations, this statistic shows that the cameras in various formations will maintain reasonable non-zero spatial angles between each other. Therefore, their camera views are less likely to coincide and provide more diverse perspectives to generate a more reliable 3D triangulation.

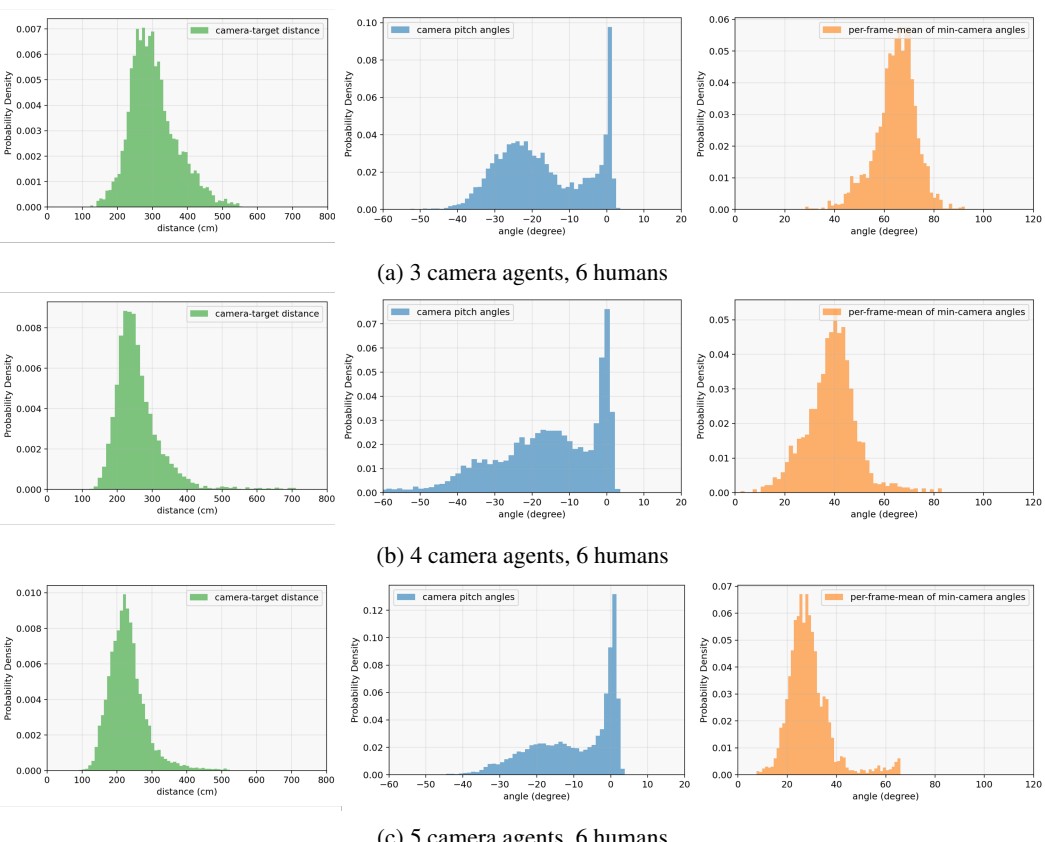

(a) 3 camera agents, 6 humans

(b) 4 camera agents, 6 humans

(c) 5 camera agents, 6 humans

Figure 15: Histograms of the three metrics for 3-, 4- and 5-camera methods. **Left Column:** *pdf* of camera-to-target distance. **Middle Column:** *pdf* of camera pitch angle. **Right Column:** *pdf* of per-frame-mean of min-camera angle. Detailed descriptions of each metric can be found in Section H.

