# OpenReview forum: "Proactive Multi-Camera Collaboration for 3D Human Pose Estimation"
_ICLR.cc/2023/Conference — ICLR 2023 poster_

### Official Review · Reviewer_W1S5 · 2022-10-22

**Confidence:** 3
**Correctness:** 3
**Technical Novelty And Significance:** 3
**Empirical Novelty And Significance:** 3
**Recommendation:** 6

**Clarity, Quality, Novelty And Reproducibility:**

- The paper is generally well-written and easy to read.
- The paper has good conceptual and technical novelty since it addresses a challenging new task with some new techniques such as CTCR and world dynamics learning.
- The paper would be hard to reproduce without a full-release of the training code and pre-trained model given the difficulty in reproducing multi-agent RL experiments

**Details Of Ethics Concerns:**

The proposed algorithm might not produce safe UAV trajectories, and collisions could happen between humans and UAVs.

**Strength And Weaknesses:**

**Strength:**

- The problem the paper aims to solve is interesting and challenging, i.e. reconstructing dynamic human motion in the wild where occlusions often happen and static camera formations are not sufficient to address them.
- Extensive experiments and ablation studies are done in the simulated environment, which shows the method outperforms the baselines and also validates the usefulness of the proposed techniques.
- The paper also plans to open-source the UE4-based RL environment used in the method, which would be beneficial to future research on proactive multi-camera human pose estimation.

**Weakness:**

- The method is not evaluated with safety metrics such as collision or interference with humans, which is rather important given the trajectory produced by the method could potentially harm the subjects.
- The simulation does not consider the dynamics of real-world UAVs and many of the jittery trajectories produced by the method might not be realizable by UAVs.
- A major concern of mine is that all the experiments are performed in simulation without real-world validation of the proposed approach. Given the method is potentially unsafe and the trajectories might be unrealizable, it is important to demonstrate its real-world applicability as prior work (e.g., AirCapRL) does.

**Summary Of The Paper:**

This paper presents a proactive multi-camera collaboration method for 3D multi-person human pose estimation with unmanned aerial vehicles (UAVs). The method is based on multi-agent reinforcement learning that treats cameras as agents and utilizes a world dynamics model to improve the performance of the model. Specifically, by additionally predicting the dynamics of camera agents and humans, the paper is able to extract more useful features for the control of the camera agents. Inspired by prior work, the paper also proposes a collaborative triangulation contribution reward (CTCR) that uses the shapley value to improve the credit assignment of reconstruction quality to agents. The paper builds its own RL environment based on Unreal Engine 4 (UE4) where human crowds are simulated with collision avoidance in various scenes. Experiments in this environment demonstrate the superiority of the proposed approach over the baselines. Extensive ablation studies are also carried out to validate the design of the method.

**Summary Of The Review:**

Overall, the paper presents an interesting approach to a challenging and useful problem. However, its evaluation is lacking, especially the real-world evaluation, which is very important given the safety implication and concerns in practicality.

== After rebuttal

The authors' response addressed most of my concerns, so I updated my rating.

---

> ### Author Response · Authors · 2022-11-17
> **Response to Reviewer W1S5 (1/1)**
>
> > ***Q1:** The method is not evaluated with safety metrics such as collision or interference with humans, which is rather important given the trajectory produced by the method could potentially harm the subjects.*
>
> We would like to make a correction to your comment since we have already implemented safety features into our method. We have addressed the safety-related issues in Appendix Section G, which is titled “Generating Safe and Smooth Trajectory”, in our first submitted manuscript. We have proposed a safety metric called “Min Camera-Human Distance” that measures the historical closest distance between any camera and any human over a period of time. Figure 13 compares the safety metric between three methods: the unsafe “Vanilla” method, augmented with Obstacle Collision Avoidance (OCA) module and augmented with Action Masking (AM). We believe anywhere below 0.25 m is dangerous and harmful to humans. As you can see from the figure, the “Vanilla” method without any safety features is unsafe. “Action masking” is mostly safe but not completely reliable due to a few edge cases. OCA is superior to AM since OCA resembles a feed-forward controller that overrides dangerous actions directly, which greatly mitigates the risks of unsafe edge cases. For more detailed descriptions of the safety features, please refers to Appendix Section G.
>
> ---
>
> > ***Q2:** The simulation does not consider the dynamics of real-world UAVs and many of the jittery trajectories produced by the method might not be realizable by UAVs.*
>
> To best simulate the dynamics of the real world, we have experimented with adding noises and delay (EMA) to the final controller output. Our result shows that our method is robust against up to 20% overshoot or overdamping in the control signal and robust against an exponential delay factor of 0.50. The result is provided in Table 9 of Appendix Section G in our first submitted manuscript.
>
> We have also already applied EMA[1][2] trajectory smoothing to make the trajectories much less jittery. Our method can generate smooth trajectories compared to the MAPPO baseline, as shown by the video demos provided on our [Google Site](https://sites.google.com/view/active3dpose/home#h.ff0owaj5914w). In this work we focus more on the high-level collaboration between cameras and less on the low-level control optimizations.  As a side note, we would like to point out that our method does not exclusively apply to Unmanned Aerial Vehicles. Still, it can also apply to any active vision system by simply adjusting the action space, e.g., restricting the z-axial movement to simulate ground vehicles.
>
> ---
>
> > ***Q3:** A major concern of mine is that all the experiments are performed in simulation without real-world validation of the proposed approach.*
>
> We agree that a real-world validation can further develop the significance of our work. Therefore, we are planning to deploy it in our future work. However, regarding the primary scope of this paper, it is to propose a learning algorithm to promote effective proactive multi-camera collaboration for 3D HPE in crowds, rather than developing a hardware system. Although this problem relates to a wide range of potential applications (that are not exclusive to UAVs), it has not been studied extensively in the context of MARL before. So we argue that the most feasible solution to the current stage is to perform all the experiments in simulation. Meanwhile, considering the sim2real gap, our experiments have accounted for multiple realistic factors in the simulation. For example, training models with safety features (collision avoidance), adding noise in the world dynamic, considering the delay in the control loop, and testing our methods in various scenarios with drastic differences in lighting conditions and landscapes (zero-shot sim2sim transfer).
>
> ---
>
> > ***Q4:** The paper would be hard to reproduce without a full-release of the training code and pre-trained model given the difficulty in reproducing multi-agent RL experiments*
>
> Thanks for your suggestion. In the supplementary, we already released our anonymized codebase in the zip file, including the training code, and environment API. Since the size of UE binaries and pre-trained models are quite massive, please refer to this [(link)](https://drive.google.com/drive/folders/18D7rBvof-AxB5BBkJ5ywvnoRCcWHdBfg?usp=share_link) to download them. Due to the sheer complexity of our project, different components have to be separate during the reviewing process. We are sorry for the inconvenience. Upon the official acceptance of this work, we will reorganize the codebase and release all the UnrealPose environment, pre-trained models, UE4 binaries, RLlib training scripts, and the visualization tool in an easy-to-use way. We have added an extra Reproducibility Statement Section in the main body of the paper.
>
> ---

---

> > ### Author Response · Authors · 2022-11-17
> > **References**
> >
> > ### Reference:
> >
> > [1] Rosati, S., Krużelecki, K., Heitz, G., Floreano, D., & Rimoldi, B. (2015). Dynamic routing for flying ad hoc networks. IEEE Transactions on Vehicular Technology, 65(3), 1690-1700.
> >
> > [2] Loquercio, A., Maqueda, A. I., Del-Blanco, C. R., & Scaramuzza, D. (2018). Dronet: Learning to fly by driving. IEEE Robotics and Automation Letters, 3(2), 1088-1095.

---

### Official Review · Reviewer_XxGE · 2022-10-24

**Confidence:** 2
**Correctness:** 3
**Technical Novelty And Significance:** 3
**Empirical Novelty And Significance:** Not applicable
**Recommendation:** 8

**Clarity, Quality, Novelty And Reproducibility:**

The paper is well-written.  The work is novel in the sense that it proposes a new formulation for the control and coordination of a group of non-stationary cameras for the task of 3D human pose estimation.  The work suggests a large engineering effort---from setting up a 3D environment to training a large, non-trivial reinforcement learning model.  I like the fact that the paper promises to open-source the virtual environment and the visualization software.

**Strength And Weaknesses:**

This is a well-written paper that studies an interesting problem.  The paper claim four "key" contributions; however, not all of these contributions have the same scientific or technological impact.  I suggest to divide this list into primary and secondary lists.  The key conributions, I feel, are: 1) problem setup and 2) the new reward formulation.  The engineering effort in setting up the 3D world that others can use for their own work seems to me a secondary contribution.

When speaking of active camera collaboration schemes for scene analysis and pedestrian tracking and the use of 3D worlds to study such multicamera systems, the following work comes to mind

Smart Camera Networks in Virtual Reality. Qureshi, F.Z.; and Terzopoulos, D. Proceedings of the IEEE (Special Issue on "Smart Cameras"), 96(10): 1640–1656. October 2008.

This is highly relevant to the work presented in this paper, and it should be cited in Multicamera Collaboration Section.  This work studies multi-camera control and uses a 3D world with autonomous pedestrians to develop and evaluate the camera coordination on the task of pedestrian tracking.

It may be useful to provide a more detailed caption for Figure 2.  Specifically, please define the variables not defined anywhere else.

It will be beneficial to provide the number of dimensions for each variable used in equations 1 to 4.  I feel that this will increase the clarity of the proposed approach.

In Section 3.4 what is the difference between the future position of target person and the future position of pedestrians?

I tried but failed to wrap my head around Eq. 6.  Perhaps it is possible to revise this discussion.  One possibility is to construct Eq. 6 term by term.  This equation deals with the reward for each camera, which, alongwith problem setup, I feel is one of the primary scientific contribution of this work.

I was pleased to see that the paper includes a discussion about the limitations of the proposed method.  Perhaps this discussion should be expanded.  For example, it seems that this work assumes near-perfect camera calibration, wich is not always possible to achieve in practice.  It is especially hard to maintain the calibration of a group of cameras that move around.

I was also pleased see the paragraph about the ethical consideration of using such systems.  I ask that this dicussion should be moved to the main body of the apaper.   It is too important to be relegated to the appendix.

Please use a different "bibstyle."  As it stands it is tedious to connect a citation to its reference.

The paper asserts that the group of cameras learn occlusion-avoiding anticipatory behavior.  Actually this was one of the reasons why the model was also trained on world-dynamics-learning-tasks.  Perhaps there is a way to provide some results that support this claim.

**Summary Of The Paper:**

The paper develops an active vision scheme where multiple non-stationary cameras reconfigure themselves to achieve high-quality 3D human pose estimation.  The camera control problem is cast within the multi-agent reinforcement learning framework.  The paper also presents a new reward formulation that incentivizes the cameras according to their weighted marginal contribution to the 3D human pose reconstruction quality.  This reward seems to address the multi-agent credit assignment issue.  The proposed model is trained within a 3D environment populated with lifelike pedestrians.  The model is jointly trained with world-dynamics-learning task; therefore, it exhibits anticipatory occlusion avoidance, improving the reconstruction accuracy.

**Summary Of The Review:**

This is an application paper.  It makes a number of technological contributions---the 3D environment, model, etc.  The paper also makes some scientific contributions---model setup and credit assignment strategy.  The paper will be of interest to the machine learning community.  It is particularly useful for those who are interested in multi-agent systems coordination problems where the individual agents rely upon vision as their primary perception modality.

---

> ### Author Response · Authors · 2022-11-17
> **Response to Reviewer XxGE (3/3)**
>
> ---
> > ***Q6:** I was pleased to see that the paper includes a discussion about the limitations of the proposed method. Perhaps this discussion should be expanded. For example, it seems that this work assumes near-perfect camera calibration, wich is not always possible to achieve in practice. It is especially hard to maintain the calibration of a group of cameras that move around.*
>
> That is a very good point. We have added this to the limitation section. However, we are seeing rising interest in parameter-free pose estimation models, which do not require online camera calibration and may help to walk around the aforementioned limitation. We will experiment with these models in our future work.
>
> ---
> > ***Q7:** I was also pleased see the paragraph about the ethical consideration of using such systems. I ask that this discussion should be moved to the main body of the a paper. It is too important to be relegated to the appendix.*
>
> We were unaware that the ethical statement would not be counted toward the page count. We have thus moved the ethical statement to the main text thanks to your suggestion.
>
> ---
> > ***Q8:** Please use a different "bibstyle." As it stands it is tedious to connect a citation to its reference*
>
> If you are referring to the hyper-reference not showing highlights (though still clickable), it is likely due to an issue with OpenReview. We have encountered a similar issue, but the hyperref highlighting works fine when we open the PDF file with Adobe Acrobat instead. Please try this to see if this works for you.
>
> ---
> > ***Q9:** The paper asserts that the group of cameras learn occlusion-avoiding anticipatory behavior. Actually this was one of the reasons why the model was also trained on world-dynamics-learning-tasks. Perhaps there is a way to provide some results that support this claim.*
>
> This is indeed attributed to having to train models with world-dynamics-learning (WDL) tasks. Here  [(link)](https://youtu.be/9-6NoaTu0HM ), we provided a video demo showing the behaviour mode of a 2-Camera MAPPO model trained with WDL tasks. In the latter part of this video, we can observe a few things that signal occlusion-avoiding anticipatory behaviours:
>
> - Two cameras roughly maintain a right-angle formation, with the red camera mostly flying over the humans and the green camera flying at an even level with the humans. The “yellow skeleton” denotes the predicted pose for the target person.
>
> - (00:12 ~ 00:16) Green Camera is aware that the target person (in yellow) is walking into the crowd, and another pedestrian is simultaneously walking into the camera view. The camera has anticipated a potential occlusion going to be caused by this pedestrian; therefore, it starts ascending and flies to an open area where occlusion is unlikely to happen. The red camera cooperates with the green camera throughout this transition. As the green camera moves toward the left, the red camera moves toward the right, thus clear signs of maintaining the right-angle formation.
>
> - (00:08 ~ 00:12) A similar situation happens here. The green camera anticipates that a pedestrian (predicted pose overlayed in purple) will occlude its view significantly, so the green camera flies to an open area with the help of the red camera to maintain a roughly right-angle formation.

---

> ### Author Response · Authors · 2022-11-17
> **Response to Reviewer XxGE (2/3)**
>
> ---
> > ***Q5:** I tried but failed to wrap my head around Eq. 6. Perhaps it is possible to revise this discussion. One possibility is to construct Eq. 6 term by term. This equation deals with the reward for each camera, which, along with problem setup, I feel is one of the primary scientific contribution of this work.*
>
> Figure 3 is an example of how to plug numbers into Eq.6 to compute CTCR for each of the three cameras. There is a breakdown of Eq.6 just below it. But just in case if it seems too obscure, here is a more intuitive description of Eq.6. The CTCR is incentivized by the Shapley Value, so the main idea is that the overall optimality needs to also account for the optimality of every possible sub-formation. In the context of an active HPE task, for a camera agent to receive the highest CTCR possible, its current position and view must be optimal both in terms of its current formation and any sub-formation possible.
>
> ***Note:** Game theory folks like to call a group of collaborating players a “coalition”. Here we apply the same concept but to a group of cameras, so we used the more initiative term “formation” instead.*
>
> Eq.6 can be further breakdown as follows:
> $$
> \varphi_r (i) = \sum_{S \subseteq [ n ] \setminus \{i\}} \frac{|S|! (n - |S| - 1)!}{n!} [ r (S \cup \{i\}) - r (S) ]
> $$
> $$
> \qquad \  = \frac{1}{n} \sum_{S \subseteq [ n ] \setminus \{i\}} \frac{|S|! (n - |S| - 1)!}{(n - 1)!} [ r (S \cup \{i\}) - r (S) ]
> $$
> $$
> \qquad \ = \frac{1}{n} \sum_{S \subseteq [n] \setminus \{i\}} \binom{n-1}{|S|}^{-1} [ r (S \cup \{i\}) - r (S) ] \qquad (\text{binomial coefficient } \binom{k}{n} = \frac{n!}{k! (n - k)!}, 0 \le k \le n)
> $$
> where $[n] = {1, 2, \dots, n}$ denotes the set of all cameras. $S$ denotes a formation (a subset) without camera $i$. $\binom{n-1}{|S|}$ is the number of combinations of subset $S$ (i.e., the binomial coefficient), which serves as a normalization term. $r (S)$ computes the reconstruction accuracy of the formation $S$. And $[r (S \cup \{i\}) - r( S )]$ computes the marginal improvement after adding the camera $i$ to sub-formation $S$.
> So this equation means we iterate over all possible $S$ and compute the marginal contribution of camera $i$ and average over all possible combinations of $( S, i )$.
>
> Suppose we have a 3-cameras formation, similarly shown in Figure 3. So $n=3$, the number of cameras. Let’s name these cameras (1,2,3), and let’s say we only care about Camera 1 for now. Since we are computing the average marginal contribution for Camera 1, we are looking for those formations that do not have Camera 1 because we want to see how much of an increase in performance resulted from the addition of Camera 1 to those formations. For all possible formations denoted by the set $[n]$, four formations satisfy this condition, $S\subseteq [n] \setminus \{i\} \longrightarrow S \in (\emptyset, 2, 3, 23)$.
>
> The binomial coefficient $\binom{n-1}{|S|}$ for a 2-camera sub-formation in a 3-cameras case is $\binom{2}{2}$, which makes sense because there exists only one unique combination that does not contain Camera 1, which is sub-formation $23$.
>  $r (23)$ computes the reconstruction accuracy of the formation $23$ and  $r (23 \cup \{1\})$ computes the reconstruction accuracy after adding Camera 1 to sub-formation $23$. Their difference gives us the marginal contribution of Camera 1. As we sum over all subsets $S$ of $[n]$ not containing Camera 1, and then divide by $\binom{n-1}{|S|}$ and the number of cameras $n$, we have the average marginal contribution of Camera 1 $ ( \varphi_r (1))$ to the collaborative triangulated reconstruction. Further multiplying this term by $n$, you have the $CTCR_1$ as shown in Eq. 6.

---

> ### Author Response · Authors · 2022-11-17
> **Response to Reviewer XxGE (1/3)**
>
> ---
> > ***Q1:** When speaking of active camera collaboration schemes for scene analysis and pedestrian tracking and the use of 3D worlds to study such multicamera systems, the following work comes to mind*
>
> >> *Smart Camera Networks in Virtual Reality. Qureshi, F.Z.; and Terzopoulos, D. Proceedings of the IEEE (Special Issue on "Smart Cameras"), 96(10): 1640–1656. October 2008.*
>
> > *This is highly relevant to the work presented in this paper, and it should be cited in Multicamera Collaboration Section. This work studies multi-camera control and uses a 3D world with autonomous pedestrians to develop and evaluate the camera coordination on the task of pedestrian tracking.*
>
> Very interesting; thanks for bringing this paper to light. This work is indeed a spiritual predecessor to our work in that Qureshi et al. proposed a coordination scheme for multiple PTZ cameras and tested their solution similarly in a synthetic environment. We have thus added this work to our related work section. We are very glad to see that we now have the capability to simulate active cameras and dynamic crowds thanks to the advance of simulation technologies over the 14 years.
>
> ---
> > ***Q2:** It may be useful to provide a more detailed caption for Figure 2. Specifically, please define the variables not defined anywhere else.*
>
> Sorry for the inconvenience. We have updated the caption for Figure 2.
>
> ---
> > ***Q3:** It will be beneficial to provide the number of dimensions for each variable used in equations 1 to 4. I feel that this will increase the clarity of the proposed approach.*
>
> We have added a complementary description of Eqn. 1-4 and a table showing the controller module feature tensors dimensions in Appendix Section D.2.
>
> ---
> > ***Q4:** In Section 3.4 what is the difference between the future position of target person and the future position of pedestrians?*
>
> The target person is the person that we would like to reconstruct the 3D pose, while the pedestrians are distractors serving as the sources of occlusions. These ($\hat{p}^{t+1}_\text{tgt},\hat{p}^{t+1}_\text{pd}$) are the same type of information, namely the 3D location of a specific human. However, there could be a varying amount of pedestrians in an environment but there is only one target person in each environment.

---

### Official Review · Reviewer_eqPb · 2022-10-24

**Confidence:** 4
**Correctness:** 3
**Technical Novelty And Significance:** 3
**Empirical Novelty And Significance:** 3
**Recommendation:** 6

**Clarity, Quality, Novelty And Reproducibility:**

# Clarity:
- The paper is well written.
# Quality:
- The paper is technically sound, and the effectiveness of the method is empirically demonstrated on various scenarios.

# Novelty:
- The proposed method is the first one to use multiple dynamic cameras for HPE via multi-agent reinforcement learning framework.

# Reproducibility:
- The source code of the environments used in the experiments are provided.

**Strength And Weaknesses:**

# Strength:
- The proposed method novel in using multiple (>3) active cameras to perform 3D HPE in human crowd.
- The proposed method introduces a decentralized framework via multi-agent reinforcement learning for multi-camera collaboration at different scales and scenarios (in simulated enviromnment).
# Weakness:
- The proposed approach relies on human recognition module to distinguish people in a scene. For some out-of-distribution human appearances, the current ReID module may not work, which will then hinder HPE task.
- The practicality of the algorithm seems limited. To perform 3D human pose estimation, dynamic cameras seem quite difficult to deploy in real-time.


**Summary Of The Paper:**

The paper introduces a multi-agent reinforcement learning approach with a Collaborative Triangulation contribution the credit assignment by using their weighted average marginal contribution for 3D human pose estimation in dynamic human crowds. The model is trained with multiple world dynamics learning tasks. The method is evaluated in four photo-realistic UE4 environments. The results show the proposed method outperforms the fixed and active baselines in different scenarios with various numbers of cameras and humans.

**Summary Of The Review:**

The paper introduces a novel dynamic multi-camera collaboration framework via multi-agent reinforcement learning to solve the 3D Human pose estimation problem. The paper is clearly written and well-structured with strong empirical evaluation in various simulated scenarios and conditions. One potential drawback of the proposed method is the practicality of the dynamic multi-camera deployment in real-world environments.

---

> ### Author Response · Authors · 2022-11-17
> **Response to Reviewer eqPb (1/1)**
>
> ---
> > ***Q1:** The proposed approach relies on human recognition module to distinguish people in a scene. For some out-of-distribution human appearances, the current ReID module may not work, which will then hinder HPE task.*
>
> This has been identified in our limitation section. It is a drawback of our current implementation, but it is not the drawback of our method. We used pre-scan appearance memory because of the ease of implementation. We believe this is a reasonable simplification as the ReID is not the main focus of our work. Our method can practically work with any ReID module that provides accurate identifications. According to our literature study, the latest ReID algorithms [1][2][3] have attained significant progress in real-time efficiency and cross-domain generalizability, which ensure practicality in real-life deployment.
>
> ---
>
> > ***Q2:** The practicality of the algorithm seems limited. To perform 3D human pose estimation, dynamic cameras seem quite difficult to deploy in real-time.*
>
> Our solution can run in real-time. The table below reports the inference time for each module of the proposed Active3DPose pipeline. The average time data are generated on a computing node with one NVIDIA RTX 2080TI GPU and one Intel Xeon E5-2699A CPU. These data are from testing our 5-camera model over 2500 environment steps in the Blank Environment. The total inference speed of our current implementation is around 8 FPS.
>
> |  | **Average Time (ms)** |
> |---|---|
> | Human Detection | 20.05 |
> | ReID Module | 28.88 |
> | 2D Pose Estimation | 31.16 |
> | 3D Triangulation | 1.43  |
> | Observation Processing| 3.97 |
> | RL Model Inference | 40.76 |
> | **Total Inference Time** | 126.25 |
>
> ---
>
> > ***Q3:** One potential drawback of the proposed method is the practicality of the dynamic multi-camera deployment in real-world environments.*
>
> Indeed, we plan to perform real-world validation as a part of our future work. In the comments above, we have addressed your potential concerns regarding the model run-time and technical dependencies. We added noises and delay (EMA) to our model and found little disturbance to the reconstruction accuracy. We also evaluated our model in different scenarios with drastic differences in lighting conditions and landscapes, where the evaluation results suggest a good robustness of our model across all environments. The main focus of this work is to propose an algorithm that promotes better collaboration between multiple cameras in an active 3D HPE task in a dynamic human crowd. This problem has not been extensively studied in the context of multi-agent RL.
>
> ---
>
> ### Reference
> [1] Ni, H., Song, J., Luo, X., Zheng, F., Li, W., & Shen, H. T. (2022). Meta Distribution Alignment for Generalizable Person Re-Identification. In Proceedings of the IEEE/CVF Conference on Computer Vision and Pattern Recognition (pp. 2487-2496).
>
> [2] Zhao, Y., Zhong, Z., Yang, F., Luo, Z., Lin, Y., Li, S., & Sebe, N. (2021). Learning to generalize unseen domains via memory-based multi-source meta-learning for person re-identification. In Proceedings of the IEEE/CVF Conference on Computer Vision and Pattern Recognition (pp. 6277-6286).
>
> [3] Dai, Y., Liu, J., Sun, Y., Tong, Z., Zhang, C., & Duan, L. Y. (2021). Idm: An intermediate domain module for domain adaptive person re-id. In Proceedings of the IEEE/CVF International Conference on Computer Vision (pp. 11864-11874).

---

### Official Review · Reviewer_FtmL · 2022-10-24

**Confidence:** 3
**Correctness:** 3
**Technical Novelty And Significance:** 3
**Empirical Novelty And Significance:** 3
**Recommendation:** 6

**Clarity, Quality, Novelty And Reproducibility:**



### Clarity
The paper is clearly presented, I also appreciate that plenty of details are given.

### Novelty
As also mentioned above, I found it quite novel to use multi-agent reinforcement learning for multi-camera motion capture.

### Reproducibility
Though I am not confident that the work can be reproduced 100% without available implementation as a reference, overall I am satisfied with the amount of details given.


**Details Of Ethics Concerns:**

None to the best of my knowledge.

**Strength And Weaknesses:**



# Strength
- The methodology is clearly motivated, and I found it quite novel to use multi-agent reinforcement learning for multi-camera motion capture.
- The method is extensively evaluated with plenty of experiments and analysis.
- The proposed virtual environment also seems a very useful tool, and can facilitate future research in the direction if this can be open-sourced.

# Weakness
Though it's a very interesting work, I am mainly concerned with two aspects: generalizability, and the understanding of the learned policy seems limited.
- **Generalizability**: as the model training seems only possible with virtual environments, and the real world can exhibit significant gaps with such environments, e.g. different body motion, different type of occlusion etc, it's not clear whether the learned policy can be generalizable to real environments. The experiments in Fig7 are helpful, I think it would be better to also add an upperbound model ("ours" trained/finetuned in the same environment) to get a sense of how the model is impacted by the domain gap. Nevertheless, I think the experiment only provides a partial view, still more evidence (e.g. experiments on real datasets) may be needed to validate the generalizability.
- **Understanding of the learned policy is limited**: I appreciate the amount of experiments provided. But still it is not fully clear what is actually learned by the policy and why it is better than the baseline. It would be helpful if authors can provide more statistics/analysis on the behavior mode of the agents and visualizations. Besides, it would also be useful to provide experiments with more cameras until the performance gap between fixed cam/learned policy is closed. (I would expect with more cameras, it will spread out to cover most views, thus the improvement from a learner polocy can become marginal)

# Other questions
I have also several minor questions, would be good if the authors can discuss them:
- wrt 3.1 Action Space, why rotation is limited to 2D and there is no roll?
- wrt fig6, there is a sudden increase in error for MAPPO at 5-6 human, it would be interesting to understand why this happens, and to provide some illustration to demonstrate the error mode. Also, would be useful to provide the same visualization for MAPPO + RLPred to understand why RLPred is so helpful in such cases.
- wrt 3.2 Perception Module, it can be helpful to provide an ablation study on used 2D pose models, and see how the errors in 2D pose would impact the learned policy. Currently it's not very clear whether the policy learned to encourage collaboration among cameras, or it's more encouraging camera to be at a view more optimal for 2D pose model. Would be interesting to provide some results with GT 2D pose (from virtual environment), and some cross results to test the generalization ability (e.g. learned from GT 2D pose, test with yolo v3 etc.)



**Summary Of The Paper:**

This paper describes a method for active (cameras can move proactively) multi-camera motion capture. The problem is formulated under multi-agent reinforcement learning framework, where the authors propose a new reward (Collaborative Triangulation Contribution Reward) to  incentivize agents according to their weighted average marginal contribution to the 3D reconstruction. The proposed method is evaluated on UE4 environments, where the method has demonstrated notable improvements over baseline, and competing methods.


**Summary Of The Review:**



Overall it's an interesting work. The methodology is clearly motivated, and I found it quite novel to use multi-agent reinforcement learning for multi-camera motion capture. Also, the method is extensively evaluated with plenty of experiments and analysis.

I have some concerns regarding generalizability of the method, and that the understanding of the learned policy seems limited. But none of my concerns are deal-breakers and I hope to discuss those in the rebuttal.

Overall I believe it's a decent contribution, and I would recommend it for acceptance.

---

> ### Author Response · Authors · 2022-11-17
> **Response to Reviewer Ftml (4/4)**
>
> ---
>
> > ***Q7:** wrt 3.2 Perception Module, it can be helpful to provide an ablation study on used 2D pose models, and see how the errors in 2D pose would impact the learned policy. Currently it's not very clear whether the policy learned to encourage collaboration among cameras, or it's more encouraging camera to be at a view more optimal for 2D pose model. Would be interesting to provide some results with GT 2D pose (from virtual environment), and some cross results to test the generalization ability (e.g. learned from GT 2D pose, test with yolo v3 etc.)*
>
> We provide an experiment showing a comparison between the 2D pose estimation model and GT.
>
> | **Env: Blank Unit: MPJPE (mm)** | **Trained with GT/ Test with GT** | **Trained with GT/ Test with Pred2D** | **Trained with Pred2D/ Test with Pred2D** |
> |:---:|---|---|---|
> | **4 Cameras** | 53.0 | 55.3 | 54.7 |
> | **5 Cameras** | 50.6 | 52.9 | 51.6 |
>
> We obtained an even better evaluation result when training and testing our models on GT. In this case, there isn’t a 2D pose model; therefore, the collaboration strategy learned by our model will not depend on the optimality of the 2D pose model. That means our method indeed learns a better collaboration for better performance instead of just hacking the 2D pose model for better accuracy.
>
> Because 3D HPE requires a strong collaboration between the agents to generate an accurate reconstruction, the local optimum for a 2D view does not imply the global optimum for the 3D triangulated pose.
>
> For example, suppose we have two camera agents with GT 2D poses as the inputs, but their positions coincide in the 3D space, or complete mirroring (180 degrees). We still cannot get an accurate reconstruction of 3D poses via triangulation from these two views [1]. This is because an accurate triangulation does not only depend on the accurate 2d poses from all cameras but also depends on their viewing angles and relative position to each other. So a better collaboration that leads to better coordination and formations is essential to generating accurate 3D reconstruction.
>
> The GT/Pred transfer shows a moderate drop in performance. To accommodate the generalizability of our RL model, we used an estimated human 3D centre as the input to the model instead of directly using the estimated 3D joints. Therefore, our model is less overfit and less susceptible to errors in 2D pose model predictions.
>
> ---
> ### Reference:
> [1] Rahimian, P., & Kearney, J. K. (2015, November). Optimal camera placement for motion capture systems in the presence of dynamic occlusion. In Proceedings of the 21st ACM Symposium on Virtual Reality Software and Technology (pp. 129-138).

---

> ### Author Response · Authors · 2022-11-17
> **Response to Reviewer Ftml (3/4)**
>
> ---
>
> > ***Q4:** Besides, it would also be useful to provide experiments with more cameras until the performance gap between fixed cam/learned policy is closed. (I would expect with more cameras, it will spread out to cover most views, thus the improvement from a learner polocy can become marginal)*
>
> In search of a topology of a fixed-camera system that could outperform our 5-camera model, we have designed six more 3D topologies that contain 8 to 24 cameras [(link to figures)](https://sites.google.com/view/active3dpose/home#h.n8dx178qwvoj). The table below shows the MPJPE (mm) evaluated from twenty 500-steps episodes (10,000 steps in total).
>
> |  | **MPJPE (mm)** |
> |---|---|
> | **6 Cams - Hexagon** | 88.0 |
> | **8 Cams - Octagon** | 78.5 |
> | **16 Cams - Double Octagon** | 86.7 |
> | **16 Cams - Hexadecagon** | 81.7 |
> | **16 Cams - Double Octagon (OH)** | 94.9 |
> | **24 Cams - "Xmas Tree"** | 87.0 |
> | **Ours, 5 Cams, active cameras** | 51.6 |
>
> Although these topologies have significantly more cameras than our 5-active-camera model, the fixed-camera approach cannot close the performance gap between fixed-camera systems and the learned policy. We also observe inversions in MPJPE when increasing the number of cameras from 8 to 16 to 24. These results suggest that the fixed-camera system has saturated the reconstruction accuracy beyond 8 cameras, and there are no further upsides to increasing the scale. We reason this is due to the errors and biases in the 2D pose model that subsequently affect the triangulation algorithm's effectiveness.
>
> ---
>
> > ***Q5:** wrt 3.1 Action Space, why rotation is limited to 2D and there is no roll?*
>
> Roll is defined as a rotational movement along the camera's optical axis. Contrary to yaw and pitch, controlling roll yields little benefit because it barely changes the amount of visible content for a camera view. And in reality, camera stabilizers on drones also rarely consider extensive control of the roll parameter for a similar reason.
> But it is certainly possible to control the camera’s roll in our UnrealPose environment; we just assume a non-variable roll in our problem formulation.
>
> ---
>
> > ***Q6:** wrt fig6, there is a sudden increase in error for MAPPO at 5-6 human, it would be interesting to understand why this happens, and to provide some illustration to demonstrate the error mode. Also, would be useful to provide the same visualization for MAPPO + RLPred to understand why RLPred is so helpful in such cases.*
>
> Here [(link)](https://sites.google.com/view/active3dpose/home#h.kwmvlpto3qlr) are video demos of the MAPPO model and MAPPO+RLPred model. The video demo of the MAPPO model helps to visualize the error mode that likely has resulted in a decrease in reconstruction accuracy; that is, the camera agents greedily adjust their positions based on individual optimality but disregard their positions relative to their collaborators. The video demo shows that two cameras controlled by the MAPPO model have adopted an orbiting strategy around the target person, which seems a viable strategy for an individual camera to avoid occlusion. But there are many instances shown in the video suggesting that their camera views may have a large overlap. For example, at around 00:07, the red camera is flying behind the green camera, which causes their views to have a large overlap and consequently leads to a bad triangulation. These are the outcomes of making manoeuvres without considering the collaborator's position and the team's optimality.
>
> In the MAPPO+RLPred video, two camera agents keep a right-angle formation and move in sync relative to the target person. We can see that when one of the camera agents moves, it clearly considers the position of its collaborator which is apparent from the behaviour mode of keeping the right-angle formation. We reason this is because the RLPred guides the model to learn to predict the dynamics of the other cameras, hence gaining the capability to anticipate the behaviour of the collaborators. RLPred also guides the model to learn the dynamics of the reward function. So if an agent understands the behavioural mode and the interest of its collaborators, the agent can act accordingly to maximize the efficiency of their collaboration.
>
> We also provide additional metrics for a more comprehensive analysis. The results in the table below suggest that 2-Cam MAPPO+RLPred has a higher Visibility Ratio (2D) and Visibility Ratio (3D), so training with WDL in the 2-camera case will lead to better occlusion avoidance.
>
> These metrics are defined as follows:
> - Visibility Ratio (2D): average percentage of visible joints over the total number of joints per camera view
> - Visibility Ratio (3D): average percentage of visible joints that are at least visible to 2 cameras over the total number of joints
>
> |  | **2-Cam MAPPO** | **2-Cam MAPPO+RLPred** |
> |---|---|---|
> | **Visibility Ratio (2D)** | 0.86 | 0.96 |
> | **Visibility Ratio (3D)** | 0.75 | 0.93 |

---

> ### Author Response · Authors · 2022-11-17
> **Response to Reviewer Ftml (2/4)**
>
> ---
> > ***Q3:** Understanding of the learned policy is limited: I appreciate the amount of experiments provided. But still it is not fully clear what is actually learned by the policy and why it is better than the baseline. It would be helpful if authors can provide more statistics/analysis on the behavior mode of the agents and visualizations.*
>
> Here on our Google Site [(link)](https://sites.google.com/view/active3dpose/home#h.4n2l7ofygd89), we provide statistics and analysis on the behaviour mode of the agents controlled by our 3-, 4- and 5-camera policies, respectively. We are interested in understanding the characteristics of the emergent formations learned by our model. Hence we proposed three quantitative measures to understand the topology of the emergent formations: (1) the min-angle between the cameras’ orientation and the per-frame-mean of the min-camera angle, (2) the camera’s pitch angle, and (3) the camera-human distance.
> The first metric might sound obscure to some readers, hence we provide rigorous definitions as follows:
> $$
> \operatorname{min-camera angle} (i) =\min_{j \ne i} \langle \text{axis of camera $i$} , \text{axis of camera $j$} \rangle
> $$
> $$
> \text{per-frame-mean of min-camera angle} = \frac{1}{n} \sum_{i \in [n]} \operatorname{min-camera angle} (i)
> $$
>
> In simpler terms, $\operatorname{min-camera angle}(i)$ finds the minimum angle between camera $i$ and any other camera $j$. “per-frame-mean of min-camera angle” is the mean of $\operatorname{min-camera angle}(i)$ for all camera $i$ in one frame. A positive camera’s pitch angle means looking upward, and a negative camera’s pitch angle means looking downward. The camera-human distance measures the distance between the target human and the given camera.
>
> Regarding the distance between cameras and humans [(link)](https://sites.google.com/view/active3dpose/home#h.4n2l7ofygd89), the camera agents actively adjust their position relative to the target human. The cameras learn to keep a camera-human distance between 2m~3m. It is neither too distant from the target human nor too close, violating the safety constraint. The camera agents surround the target at an appropriate distance to have a better resolution on the 2D views. In the meantime, as the safe-distance constraint is enforced during training, the camera agents are prohibited from aggressively minimizing the camera-human distance.
>
> The histograms [(link)](https://sites.google.com/view/active3dpose/home#h.4n2l7ofygd89) for the camera’s pitch angle suggest that the cameras mostly maintain negative pitch angles. Their preferred strategy is to hover over the humans and capture images at a higher altitude. This is likely because of emergent occlusion avoidance. The cameras also emerge to fly at an even level with the humans (where the pitch angle approximately equals 0) to capture more accurate 2D poses of the target human. This propensity is apparent from the peaks at x=0 angles in the histograms. A relatively wide distribution of the pitch angle histogram suggests that the camera formation is spread out in the 3D space and dynamic adjustments of flying heights and pitch angles by the camera agents.
>
> For the average angle between the cameras’ orientations [(link)](https://sites.google.com/view/active3dpose/home#h.4n2l7ofygd89), this statistic shows that the cameras in various formations will maintain reasonable non-zero spatial angles between each other. Therefore, their camera views are less likely to coincide and provide more diverse perspectives to generate a more reliable 3D triangulation.

---

> ### Author Response · Authors · 2022-11-17
> **Response to Reviewer Ftml (1/4)**
>
> ---
> > ***Q1**: Though it's a very interesting work, I am mainly concerned with two aspects: generalizability and the understanding of the learned policy seems limited.
> Generalizability: as the model training seems only possible with virtual environments, and the real world can exhibit significant gaps with such environments, e.g. different body motion, different types of occlusion etc., it's not clear whether the learned policy can be generalizable to real environments.*
>
> Experimenting with real equipment and human-machine interactions are within our plan for future work. Currently, we have simulated the noisy conditions that we are going to encounter during real-life deployment, namely:
> - (1) stochasticity in human behaviours.
> - (2) variations in visual observations like various lighting conditions and landscapes.
> - (3) noisy and delayed control signal
>
> In response to the first issue (different body motions and different types of occlusion): during training, we simulate humans with different walking dynamics, i.e. random walking speed and different walking animations accordingly to cover as many types of body motion as possible. We also simulate humans with stochastic trajectories and different body types. These augmentations can lead to various types of occlusion and reduce the sim2real gap. This is a common technique used by many RL works to alleviate the sim2real transfer issue.
>
> For the second issue (variations in visual observations): we adopted publicly-available pre-trained CV models as a part of our observation processing module to handle visual variations in different scenes. Empirical results in different virtual environments demonstrate that our method can transfer without much degradation to the reconstruction accuracy.
>
> To accommodate the third issue (generalize to real environments), we have added noises and delays to the final output of an agent’s control policy to simulate the real-world dynamics. We have simulated up to 20% fluctuations in output values and an exponential delay factor of 0.5, which we have found little impact on the reconstruction accuracy of our solution. It demonstrates the robustness of our method in a simulated real-world deployment. For more details, please refers to Appendix Section G.
>
> ---
> > ***Q2:** The experiments in Fig7 are helpful, I think it would be better to also add an upperbound model ("ours" trained/finetuned in the same environment) to get a sense of how the model is impacted by the domain gap.*
>
> Thanks for your suggestion. We have added the experiment results for the upper-bound models from two test environments: SchoolGym and UrbanStreet. These are more complex environments than the Blank environment, and directly trained in these environments will help to push model performance further. The results show that the 3-camera models exhibit the greatest domain gap between different environments, and the generalizability is better for 4- and 5-camera models.
>
> Below are the results of the upper-bound models of the SchoolGym and UrbanStreet environments:
>
> |  | Trained on Blank / Test on SchoolGym | Trained & Test on SchoolGym |
> |---|---|---|
> | 3 Cameras | 99.4 | 74.3 |
> | 4 Cameras | 67.7 | 65.1 |
> | 5 Cameras | 64.6 | 59.6 |
>
> |  | Trained on Blank / Test on UrbanStreet | Trained & Test on UrbanStreet |
> |---|---|---|
> | 3 Cameras | 103.6 | 71.3  |
> | 4 Cameras | 66.4 | 55.9 |
> | 5 Cameras | 61.9 | 51.9 |

---

### Author Response · Authors · 2022-11-17
**General Comment and Revision Note**

### General Comments:
We thank all reviewers for their time and effort spent reviewing our work. We apologize for our delayed responses in the discussion phase. We hope that our rebuttal responses have addressed your concerns. Should you have any other questions, please feel free to discuss with us.

### Revision Notes:
- We improved the writing of the problem formulation to make it more rigorous. Specifically, we distinguish between the camera’s local view $o_i$ and the agent’s local observation $\tilde{o}_i$.
- We added Appendix Section D.2 and an Ethics Discussion on Page 10 in response to the request by Reviewer XxGE.
- We have updated the related work and the limitation sections in response to the requests by Reviewer XxGE.
- Figure 2 on Page 5 is added with more descriptive texts in response to the request by Reviewer XxGE.
- We added a Reproducibility Statement on Page 10 in response to the comments by Reviewer W1S5.

Other than the items above, a few suggestions by the reviewers are indeed very helpful in extending the comprehensiveness of this work. Therefore, we have added the tables, figures and analysis produced during the rebuttal discussion period to the Appendix of our manuscript. These include Section D.5, Table 5, Figure 13, Table 10, Section G and Section H, Figure 15.

Everything we changed since the first submitted manuscript has been marked in blue.

---

### Decision · Program_Chairs · 2023-01-20

**Decision:**

Accept: poster

**Justification For Why Not Higher Score:**

The paper does not address the issue of generalization to real environments, but just adds noises and delays to the final output of an agent’s control policy to simulate the real-world dynamics.

**Justification For Why Not Lower Score:**

The method of multi-agent for multi-camera based motion capture is novel.

**Metareview: Summary, Strengths And Weaknesses:**

This work introduces a new method for multi-camera-based motion capture. The authors designed a multi-agent reinforcement learning architecture with a new Collaborative Triangulation Contribution Reward. The method is tested on UE4 environments, and achieves good results. All the four reviewers are positive on this paper, and points out that the method is novel. Reviewers also pointed out some weaknesses, e.g., the generalization ability to real environments is not well addressed - while authors mention this will be a future work. Overall, the paper is interesting and results are promising. Yet, the authors still need to carefully improve the final version based on the comments of reviewers, which definitely are quite helpful.

**Note From Pc:**

if the above contains the word "oral" or "spotlight" please see: "oral" presentation means -> notable-top-5% and "spotlight" means -> notable-top-25%. As stated in our emails, we are disassociating presentation type from AC recommendations